# Non-coding RNA mediates the defense-associated reverse transcriptase (DRT) anti-phage oligomerization transition

Jie Han [1,2,3,9], Bin Liu[1,2,9], Jingjing Tang [4,5,9], Shuqin Zhang[1,2,9], Xiaoshen Wang [1], Xuzichao Li [6], Qian Zhang [1], Zhikun Liu [1], Wanyao Wang[7], Yingcan Liu [1], Ruimin Zhou[1], Hang Yin[1], Yong Wei[6], Zhuang Li [8], Minjie Zhang [7✉], Zengqin Deng [4✉] & Heng Zhang [1✉]

## Abstract

**Defense-associated reverse transcriptase (DRT) systems are implicated in prokaryotic resistance to viral infections, yet the molecular mechanisms underlying their functionality remain largely unknown. Here, we characterize a two-component DRT9 system, composed of a reverse transcriptase (RT) and a non-coding RNA (ncRNA), which exhibits a protein-primed DNA synthesis activity upon phage infection. We also determine its cryo-electron microscopy (cryo-EM) structures in different functional states. DRT9 RT binds to ncRNA, forming a dimer of dimers configuration that assembles into a trimer of dimers upon substrate binding. This oligomerization transition, crucial for DRT9-mediated anti-phage defense, is facilitated by a ncRNA cooperative self-assembly manner. Furthermore, substrate binding induces large conformational movements around the catalytic pocket of DRT9 RT, revealing a "lock-switch" mechanism for enzymatic activation. Notably, phylogenetic analysis and functional assays identify a unique N-terminal helix extension required for ncRNA stabilization and enzymatic activity, distinct from previously reported reverse transcriptase systems. Overall, our findings illuminate the molecular basis of DRT9-mediated antiviral defense and expand the functional and mechanistic diversity of the DRT family.**

**Keywords** Reverse Transcriptase; DRT9; Non-coding RNA; Protein-Primed cDNA Synthesis; Bacterial Antiviral Defense
**Subject Categories** Microbiology, Virology & Host Pathogen Interaction; RNA Biology; Structural Biology

## Introduction

The evolutionary dynamics between bacteria and phages have driven the diversity of prokaryotic defense systems (Georjon and Bernheim, 2023; Hampton et al, 2020; Koonin et al, 2020; Rousset and Sorek, 2023). Among these, prokaryotic reverse transcriptases (RTs) have evolved distinct strategies of anti-phage immunity through cDNA synthesis (Bobonis et al, 2022; Kojima and Kanehisa, 2008; Moelling et al, 2017). Although the anti-phage activity of RT enzymes was first hypothesized in the 1990s (Xiong and Eickbush, 1990), the detailed molecular mechanisms underlying most RT-based defense systems remain unresolved, with only a few members characterized. Prokaryotic RTs comprise diverse groups of lineages, such as retron RTs, CRISPR-Cas-associated RTs, "unknown group" (UG) RTs, and abortive infection (Abi)-like RTs (Gonzalez-Delgado et al, 2021; Millman et al, 2020; Simon and Zimmerly, 2008; Toro et al, 2019; Zimmerly and Wu, 2015). Systematic classification categorizes UG/Abi RTs into three classes: Class 1 UG/Abi RTs fused with HEAT-like repeats, Class 2 members featuring a single RT domain, and Class 3 RTs commonly associated with nitrilase or phosphohydrolase domains (Mestre et al, 2022). Notably, some UG/Abi RTs have been experimentally demonstrated to combat phage infection and are thus designated as defense-associated RT (DRT) family. For instance, DRT2 system, a Class 2 UG/Abi member, thwarts phage infection by generating toxic proteins through RT-synthesized repetitive cDNA transcripts from non-coding RNA (ncRNA) template (Gao et al, 2020; Tang et al, 2024; Wilkinson et al, 2024).

Like DRT2, DRT9 belongs to the Class 2 UG/Abi lineage, consisting of an RT protein and a ncRNA, but possesses distinct sequence and structural features compared to the DRT2 system. Here, we biochemically characterized the DRT9 system and found

[1]Key Laboratory of Immune Microenvironment and Disease (Ministry of Education), State Key Laboratory of Experimental Hematology, Tianjin Medical University Cancer Institute and Hospital, The Province and Ministry Co-sponsored Collaborative Innovation Center for Medical Epigenetics, Tianjin Institute of Immunology, School of Basic Medical Sciences, Tianjin Medical University, Tianjin, China. [2]Tianjin Key Laboratory of Cellular Homeostasis and Disease, Department of Biochemistry and Molecular Biology, Tianjin Medical University, Tianjin, China. [3]Department of Anatomy, School of Basic Medical Sciences, Tianjin Medical University, Tianjin, China. [4]Key Laboratory of Virology and Biosafety, Wuhan Institute of Virology, Chinese Academy of Sciences, Wuhan, China. [5]University of Chinese Academy of Sciences, Beijing, China. [6]The Cancer Hospital of the University of Chinese Academy of Sciences (Zhejiang Cancer Hospital), Institute of Basic Medicine and Cancer (IBMC), Chinese Academy of Sciences, Hangzhou, China. [7]Department of Bioinformatics, School of Basic Medical Sciences, Tianjin Medical University, Tianjin, China. [8]State Key Laboratory of Biocatalysis and Enzyme Engineering, School of Life Sciences, Hubei University, Wuhan, China. [9]These authors contributed equally: Jie Han, Bin Liu, Jingjing Tang, Shuqin Zhang. ✉E-mail: zhangmj@tmu.edu.cn; dengzengqin@wh.iov.cn; zhangheng134@gmail.com

that it preferentially utilizes dATP to synthesize poly(dA) products for phage defense. Using cryo-electron microscopy, we further determined the structures of DRT9 in distinct functional states. Structural analyses revealed that substrate-free DRT9 assembles as a dimer-of-dimers (tetrameric assembly), which undergoes an oligomeric reorganization into a trimer-of-dimers (hexameric assembly) upon substrate binding. Moreover, diverging from canonical Abi RT systems that lack ncRNA components and depend solely on protein-protein interactions for self-oligomerization (Figiel et al, 2022; Gapinska et al, 2024), the ncRNA component promotes DRT9 oligomerization in both substrate-free and bound states. Two structural elements, an arginine-finger hinge and a β-hairpin latch, cooperatively maintain the catalytic pockets in autoinhibited conformations. Substrate binding triggers concerted displacement of these elements, exposing the enzymatic active site. Meanwhile, dNTPs induce allosteric conformational changes in a flexible loop region essential for tetramer assembly, expanding the angle between two dimer units by 30°. This angular shift facilitates the transition from the tetrameric to the hexameric assembly of DRT9, further promoting its catalytic activation.

# Results

## Biochemical characterization of the DRT9 system

The UG/Abi family encompasses diverse reverse transcriptase systems, most of which are typically co-localized with other defense modules or domains in genomic loci and lack ncRNA components (Mestre et al, 2022). However, a small group of UG/Abi members, including DRT2 (UG2), DRT3 (UG3 + UG8), and DRT9 (UG28), have acquired ncRNA elements for antiviral protection (Mestre et al, 2022) (Fig. 1A, B). The DRT2 system from *Klebsiella pneumoniae* comprises a protein harboring solely the RT domain and an upstream 280 nucleotides (nt) ncRNA (Fig. 1B). The DRT2 system has been demonstrated to express a toxic protein encoded by repetitive cDNA synthesized from ncRNA (Tang et al, 2024; Wilkinson et al, 2024). Phylogenetic analysis reveals that DRT9 shares a common ancestor with DRT2 (Fig. 1A), with both systems exhibiting an RT-domain-only protein and a ncRNA, a configuration rarely observed in other UG members (Mestre et al, 2022). However, the RT proteins from DRT2 and DRT9 systems are only about 30% similar in sequence. Plaque reduction assays and growth curve analysis demonstrated the dependency of DRT9-mediated phage resistance on both catalytic RT integrity and ncRNA (Fig. 1C; Appendix Fig. S1A). This contrasts with canonical UG/Abi systems that typically require additional defense modules or domains, highlighting the evolutionary innovation of ncRNA in DRT systems and the critical role of ncRNA in DRT9-mediated antiviral function.

Although DRT9 shares a common ancestor with DRT2, the two systems exhibit distinct product profiles based on in vitro reverse transcription assays (Tang et al, 2024; Wilkinson et al, 2024) (Fig. 1D). To elucidate how the DRT9 system confers anti-phage defense through its reverse transcriptase activity, we sequenced its cDNA products generated in vitro. While the RT protein of the DRT2 system has been shown to reverse transcribe its associated ncRNA (Tang et al, 2024; Wilkinson et al, 2024), sequencing of

DRT9 cDNA products revealed no detectable alignment to its ncRNA, and only a few reads mapped to the *E. coli* genome. We further analyzed the unmapped reads and found a striking enrichment of poly-A motifs, as revealed by statistical analysis of homopolymer composition and de novo motif discovery (Fig. 1E). These results indicate that DRT9 may preferentially use dATP as substrate for reverse transcription. To further investigate this substrate specificity, we conducted reverse transcription analysis using individual dNTPs. Indeed, cDNA products were detected only in the presence of dATP (Fig. 1F; Appendix Fig. S1B). These findings indicate that the DRT9 system specifically utilizes dATP as substrate to synthesize poly(dA) sequences. DRT2 has been shown to function as a monomeric protomer (Wilkinson et al, 2024), whereas size-exclusion chromatography (SEC) and cryo-EM analysis revealed that DRT9 assembles into higher-order oligomeric complexes (Appendix Fig. S1C,D), implying a distinct anti-phage mechanism.

## DRT9 tetramer structure reveals distinct RT architecture and ncRNA fold

DRT9 system from *Escherichia coli* includes a 499 amino acids DRT9-RT protein and a 188 nucleotides ncRNA (Figs. 1B and 2A). DRT9-RT protein comprises five domains, including an N-terminal extension (NTE, residues 1–64) domain, a fingers domain (residues 65–159 and 176–211), a palm domain (residues 160–175 and 212–315), a thumb domain (residues 316–371 and 394–436) and a C-terminal extension (CTE, residues 372–393 and 437–499) domain (Fig. 2A). To elucidate the structural basis of DRT9-mediated immunity, we reconstituted and purified the DRT9-RT-ncRNA binary complex (hereafter referred to as DRT9) and determined its cryo-EM structure at a resolution of 3.49 Å (Fig. 2B,C; Appendix Fig. S2). Well-defined cryo-EM density map enabled unambiguous tracing of the majority of the RT protein and ncRNA, except for residues 494–499 of the RT and 42-45 nucleotides of the ncRNA (Fig. 2C,D; Appendix Fig. S3). The overall structure of the DRT9 complex adopts a butterfly-like architecture assembled from four DRT9 protomers. Two DRT9 protomers, designated as A/A' or B/B', associate into an antiparallel, head-to-head dimer through the fingers, palm, and CTE domains of RT protein, constituting one wing of the butterfly-like structure. Two DRT9 dimers, referred to as wing-like units, further arrange in parallel to form a shoulder-to-shoulder tetramer through fingers and palm domains (Fig. 2B,C).

Compared to the DRT2 system, the overall architecture of DRT9-RT is almost similar to the DRT2-RT, except for the NTE and fingers parts (Fig. 2E,F; Appendix Fig. S4A). A prominent N-terminal long α-helix and its following loop on the NTE domain protrude from the fingers region and point toward the thumb subdomain, resulting in a unique triangular shape for the DRT9 RT (Fig. 2E,F). Additionally, the N-terminal α1 helix is situated above the potential enzymatic pocket and makes extensive contacts with ncRNA. Particularly, the Y25 within α1 helix stacks against U121 (Fig. 2E; Appendix Fig. S4A,B). An alanine substitution at this position, with the potential to disrupt the interaction, almost abolished the reverse transcriptase activity, suggesting a regulatory role of NTE (Appendix Fig. S4C). Moreover, the secondary structure of ncRNA in DRT9 exhibits evident differences compared to that in DRT2 (Appendix Fig. S4D). The DRT9 ncRNA consists

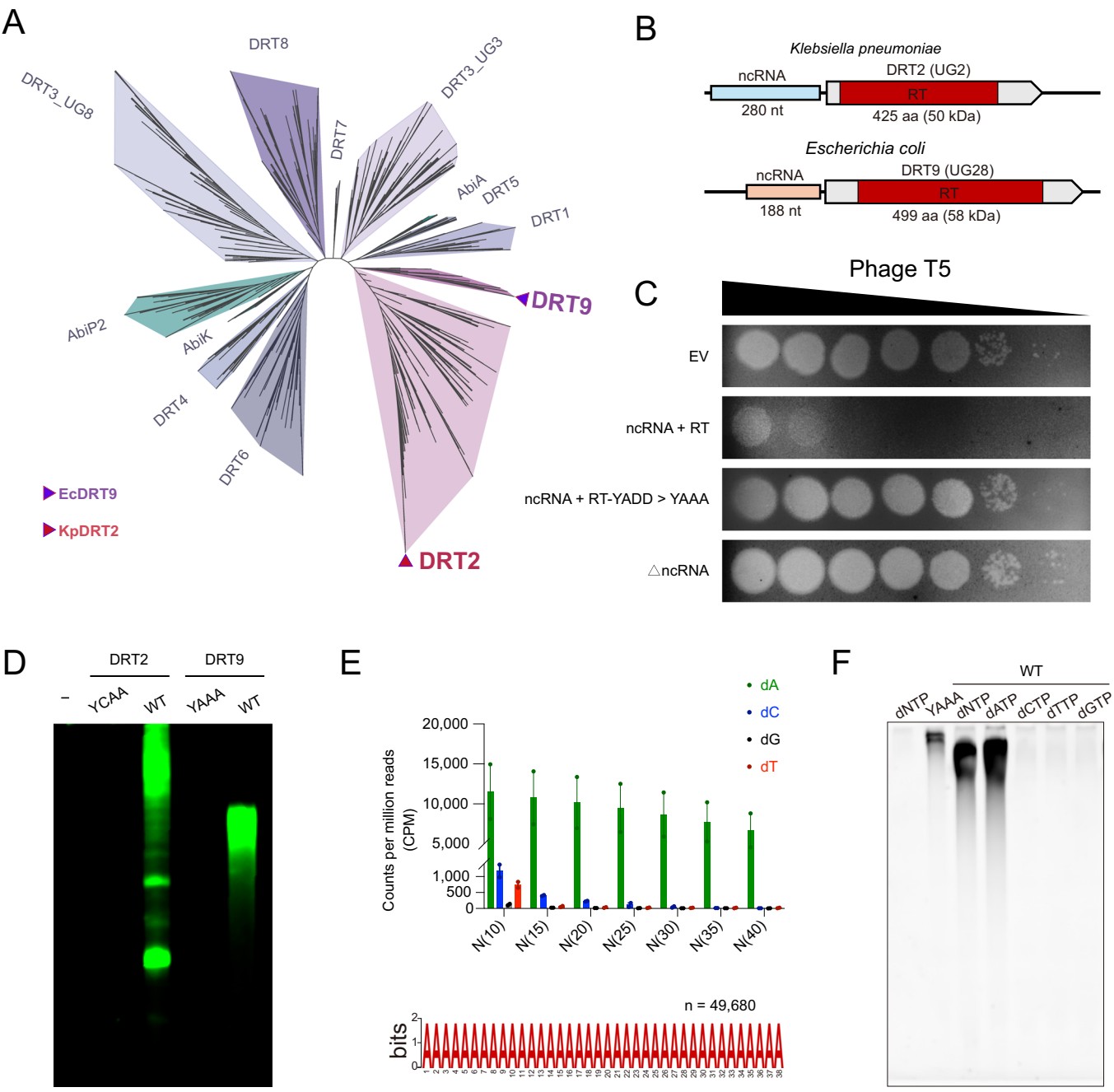

**Figure 1.  Biochemical characterizations of the DRT9 anti-phage defense system.**

(A) Phylogenetic tree of representative DRT and Abi systems. Phylogenetic studies show that the DRT9 system and the DRT2 system have a common ancestor. (B) Schematic representation of the DRT2 (UG2) system from *Klebsiella pneumoniae* and the DRT9 (UG28) system from *Escherichia coli*. Both systems comprise a reverse transcriptase (RT) and a non-coding RNA (ncRNA). (C) plaque assays with T5 phage on a lawn of BL21-AI carrying different plasmids encoding DRT9 variants. EV empty vector, RT reverse transcriptase, RT-YAAA DRT9 variant with catalytic sites mutated, ΔncRNA deletion of the ncRNA in DRT9 system. The assays are performed in triplicate. (D) In vitro reverse transcription assay of DRT2 and DRT9 systems variants. WT wild-type DRT system, YCAA DRT2 variant with catalytic sites mutated, YAAA DRT9 variant with catalytic sites mutated. The reaction products were visualized by TBE-urea PAGE. The gel is representative of three replicates. (E) The upper panel represents the normalized counts of different poly-N motifs in unmapped reads and the lower panel shows the results of motif analysis performed on unmapped reads. (F) In vitro reverse transcription assay of DRT9 system supplemented with different substrates. The gel represents three replicates. Source data are available online for this figure.

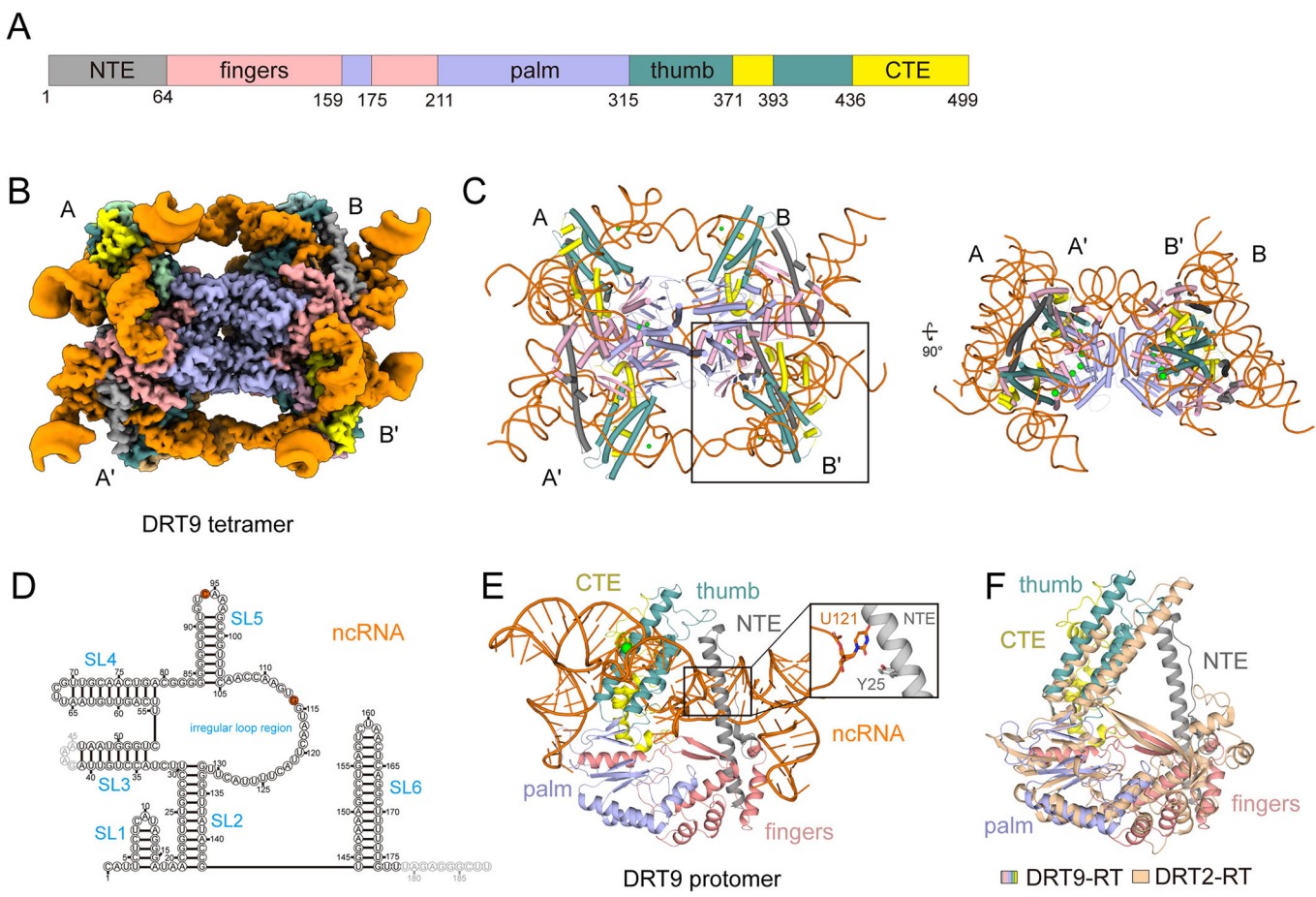

**Figure 2. Overall structure of DRT9 tetramer.**

(**A**) Domain organization of DRT9 RT protein. The domains are colored in gray (NTE, N-terminal extended), pink (fingers), medium purple (palm), deep teal (thumb) and yellow (CTE, C-terminal extension). (**B**) Cryo-EM density map of DRT9 tetramer complex. ncRNA is shown in orange. Protein domains are colored in the same scheme as in Fig. 2A. (**C**) Atomic model of DRT9 tetramer. (**D**) Schematic representation of the DRT9 ncRNA. DRT9 ncRNA has six stem loops (SLs 1-6). (**E**) Atomic model of DRT9 monomer derived from the tetramer complex. The same color scheme as Fig. 2B is applied. (**F**) Structural comparison of DRT9 RT and DRT2 RT (PDB: 9C0I). The color scheme of DRT9 RT is the same as Fig. 2A, while DRT2 RT is colored in wheat.

of 188 nucleotides and contains six stem-loops, along with an irregular loop region (A106-U131, Fig. 2D). Within the DRT9 protomer, stem-loops 1, 2, and 3, namely SL1, SL2, and SL3, wrap around the RT protein, establishing specific contacts with the fingers, NTE, and CTE domains of DRT9-RT, respectively, thereby forming the scaffold of the ncRNA (Fig. 2D,E; Appendix Fig. S5). SL4-SL6, together with the irregular loop region on the ncRNA extend outward from the DRT9 complex and show no interaction with the RT protein within the protomer. This unique ncRNA architecture, which diverges from that in the DRT2 system (Appendix Fig. S4D), suggests a specialized role in DRT9 function.

## RT- and ncRNA-mediated DRT9 tetramer assembly

The DRT9 tetramer adopts a butterfly-like structure with two parallel wings, each wing consisting of two centrosymmetric RT units arranged in a head-to-head orientation (Fig. 3A). This tetramer configuration is stabilized by dual interaction networks, including intra-wing contacts preserving integrity of dimer units and inter-wing interactions facilitating tetramerization (Fig. 3B–D;

Appendix Fig. S6A). Unlike canonical Abi RTs, such as AbiA/AbiK, which oligomerize into dimer or hexamer through RT-mediated interactions involving their helical/HEPN domains (Figiel et al, 2022; Gapinska et al, 2024), the DRT9 system incorporates ncRNA as a structural factor for tetramer assembly. The ncRNA employs dual functional roles by stabilizing individual dimer units through RNA-protein interactions and bridging adjacent wings via specialized base-pairing interactions (Fig. 3B,C; Appendix Figs. S5 and S6A).

Within each dimer, the SL4 of the ncRNA from protomer A docks onto helices α7-α8 of the RT domain in protomer A' through electrostatic complementarity. A flipped uracil (U67) within SL4 inserts into a charged pocket formed by residues R205 and K208, generating essential cation-π and hydrogen-bonding interactions for dimer assembly (Appendix Fig. S6A). Additionally, extensive protein-protein interactions involving helices α4, α5, α9, α10, and α16, together with the α7–α8 loop on the RT protein, reinforce dimer formation through polar interactions (Appendix Fig. S6A). The α10 helices and an assembly loop (residues 288-299) cooperatively mediate the side-by-side arrangement of the two

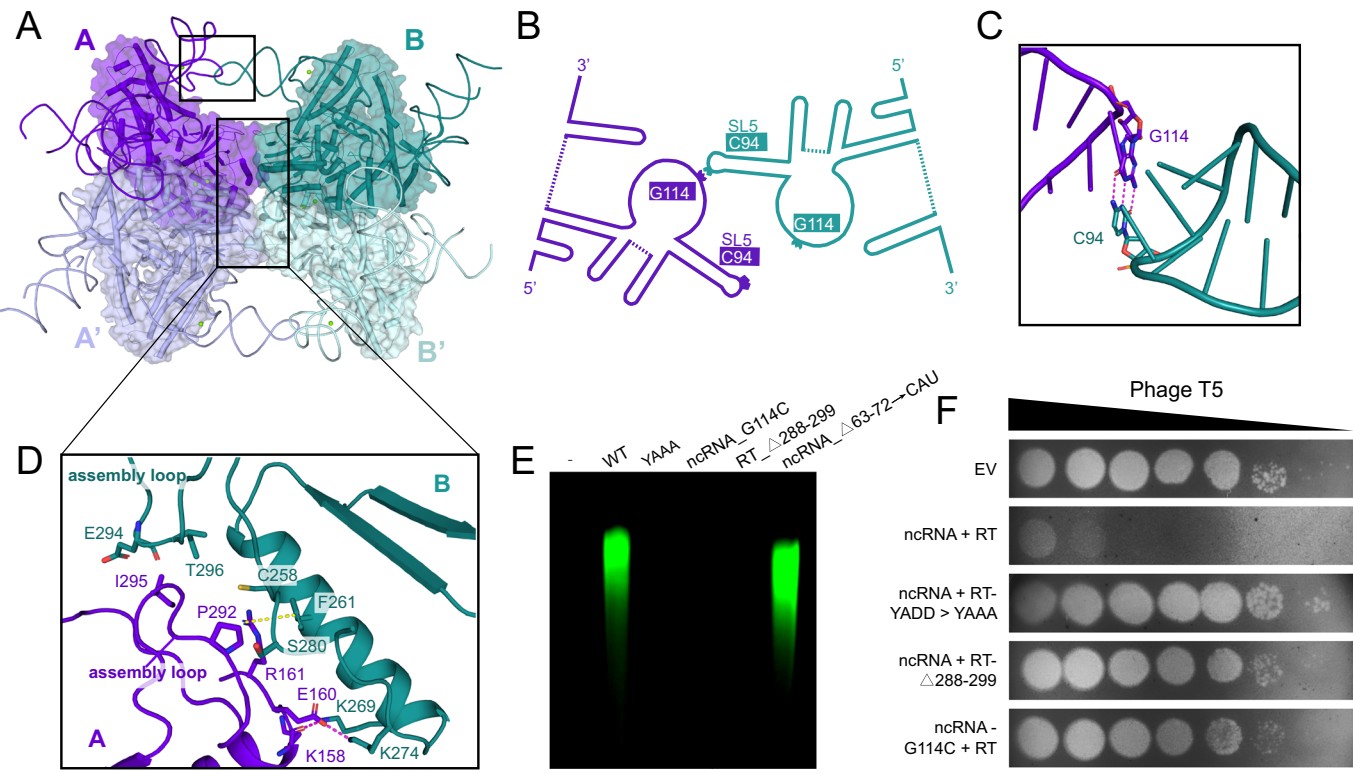

**Figure 3. DRT9 ncRNA is essential for DRT9 tetramer assembly.**

(**A**) The tetramerization of DRT9 monomers. The DRT9 monomers are colored purple (A), deep teal (B), light blue (A') and pale cyan (B'), respectively. The monomers A/A' form the DRT9 dimer unit A, the B/B' form the DRT9 dimer unit B. (**B**) Schematic diagram of ncRNA-mediated inter-unit interaction. (**C**) The C94-G114 base pair is specific for DRT9 oligomerization. (**D**) Close-up view of the inter-unit interaction surface of DRT9 RT. The key interacting residues are shown in stick representation. (**E**) In vitro reverse transcription assay of DRT9 WT and mutants. WT, wild-type DRT9; YAAA, DRT9 variant with catalytic residues mutated. The gel is representative of three replicates. (**F**) Phage plaque assays with T5 phage on a lawn of BL21-AI harboring empty vector (EV) and various DRT9 variants. The images are representative of three biological replicates. Source data are available online for this figure.

wings through polar and hydrophobic interactions (Fig. 3D). Additionally, DRT9 tetramerization is also facilitated by ncRNA-mediated inter-wing interactions. Specifically, the protruding SL5 of protomer B (or A') from one wing engages a conserved loop region in protomer A (or B') from the other wing, making C94-G114 Watson-Crick base pairs (Fig. 3B,C). The deletion of the assembly loop (Δ288-299) or the substitution of G114 with cytosine (G114C) disrupted the higher-order oligomerization of DRT9, abolished its cDNA synthesis activity, and eliminated phage resistance. This demonstrates the essential role of RT-ncRNA interactions in DRT9 function (Fig. 3E, F; Appendix Fig. S6B). Taken together, DRT9 assembles into a tetrameric complex stabilized by both RT- and ncRNA-mediated interactions, each being critical for its activity.

## Substrate-induced tetramer-to-hexamer transition mediates DRT9 activation

Reverse transcriptase uses the dNTP substrate to produce cDNA products. Intriguingly, upon incubation with dNTPs, DRT9 exhibited a slower migration in the native PAGE gel, indicating that DRT9 oligomerizes into a higher assembly upon activation (Appendix Fig. S7). We then incubated DRT9 with dNTPs and used cryo-EM to investigate this higher assembly activation state

(Fig. 4A,B). Consistent with the SEC and native PAGE results, cryo-EM data revealed "clover-shaped" hexameric structures of DRT9 except for "butterfly-shaped" tetramers based on 2D/3D classifications (Fig. 4A,B; Appendix Fig. S8). A distinct subclass of the hexameric DRT9 structures exhibited clear EM density corresponding to a bound dATP molecule, indicative of an active state. We further resolved the structure of this substrate-bound "clover-shaped" DRT9 complex at 3.46 Å resolution (Fig. 4B; Appendix Fig. S8). The high-quality EM density allowed us to clearly trace the DRT9-RT protein and ncRNA, together with a substrate molecule near the catalytic pocket.

To evaluate the structural basis of the substrate-mediated oligomer transformation and activation of DRT9, we analyzed the structures of the apo DRT9 tetramer and substrate-bound DRT9 hexamer. In apo DRT9 tetramer, DRT_A/A' and DRT_B/B' dimer units are organized in parallel through interactions between the α10 helix and the assembly loop. In the top view, the α10 helix from DRT9-A and DRT9-B' protomers are aligned in parallel, while the α10 helix of DRT9-A' and DRT9-B protomers forms a 90° inter-unit angle (Fig. 4C). This restricted space within the inter-unit angle precludes hexamer formation. In the substrate-bound DRT9 hexamer, three DRT9 dimer units assemble into a trimer of dimers, aligning side by side (Fig. 4B,D). The presence of dNTPs induces significant conformational rearrangements (Fig. 4A,B), shifting the

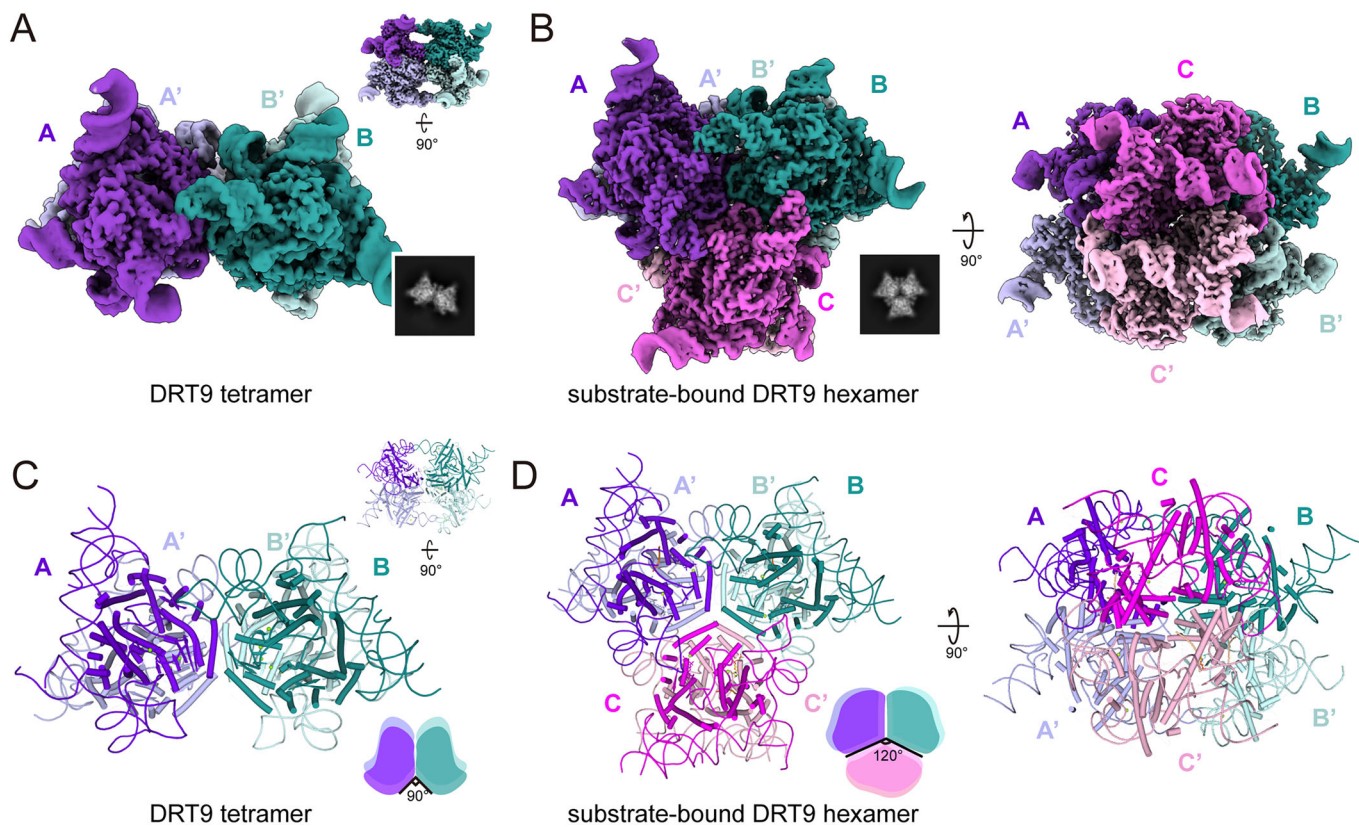

**Figure 4. The transition of the DRT9 tetramer to a hexamer induced by dNTPs.**

(A) Cryo-EM density map of DRT9 tetramer and corresponding 2D classification average. (B) Cryo-EM density map of substrate-bound DRT9 hexamer and corresponding 2D classification average. (C) Atomic model of DRT9 tetramer. The corresponding schematic is represented in the lower right corner. (D) Atomic model and corresponding schematic of substrate-bound DRT9 hexamer. The transition from tetramer to hexamer induces an obvious angle change of DRT9 dimer inter-units.

inter-unit angle from 90° to 120° (Fig. 4C,D). This angular expansion enlarges the inter-unit space, thereby facilitating hexamer assembly and further triggering DRT9 activation (Fig. 4D). Notably, the RT- and ncRNA-mediated inter-unit interactions in the DRT9-dATP hexamer closely resemble those observed in the DRT9 tetramer (Fig. 3A,B; Appendix Fig. S9). Taken together, the substrate-mediated tetramer-to-hexamer transition is a critical step for DRT9 activation, and both RT- and ncRNA-mediated inter-unit interactions are essential for the tetrameric and hexameric assembly.

## Structural basis for DRT9 enzymatic activity

Structural and sequence alignments demonstrated that DRT9 shares a conserved YXDD motif (residues 243–246) with conventional RTs, such as DRT2, AbiA, and AbiK, located on the β5-β6 loop of its palm domain (Fig. 5A,B; Appendix Figs. S10 and S11A). Notably, structural elements incorporating the YXDD motif, including the β5-β6 hairpin, are conserved across DRT and Abi systems (Fig. S10). Alanine substitutions introduced in the conserved YXDD motif, along with mutations in neighboring residues, completely abolished the enzymatic activity of DRT9, underscoring the essential role of these conserved structural elements in DRT9 catalysis (Fig. 5A,F). Adjacent to the catalytic

sites (D245, D246), a dATP molecule could be modeled into the EM density. An extra cryo-EM density was observed adjacent to the dATP molecule, which may belong to the cDNA products (Fig. 5A). Intriguingly, four nucleotides (U124–U127) in the loop region of the ncRNA, are likely to be positioned near the catalytic site, with increased flexibility upon dNTP binding (Fig. 5A). Furthermore, sequence alignment of the ncRNA reveals that the four uracil residues (U124–U127) are highly conserved across DRT9 homologs (Appendix Fig. S11B). The substitution of these uracil nucleotides with guanine led to the complete loss of DRT9-mediated anti-phage defense, underscoring their indispensable role in DRT9 activity (Appendix Fig. S11C).

## Substrate-induced activation mechanism in DRT9

To further elucidate the structural basis of DRT9 activation upon dNTP binding, we compared the substrate-free tetrameric structure with the substrate-bound hexameric structure of DRT9. The overall architecture remains similar, with differences primarily localized to the polymerization interface and the catalytic pocket (Fig. 5C–E). In detail, compared to the substrate-free state, substrate binding induces an upward movement in the assembly loop (residues 288–299) of RT protein. This shift enlarges the angle between two adjacent DRT9 dimer units from 90° to 120°, creating space for a

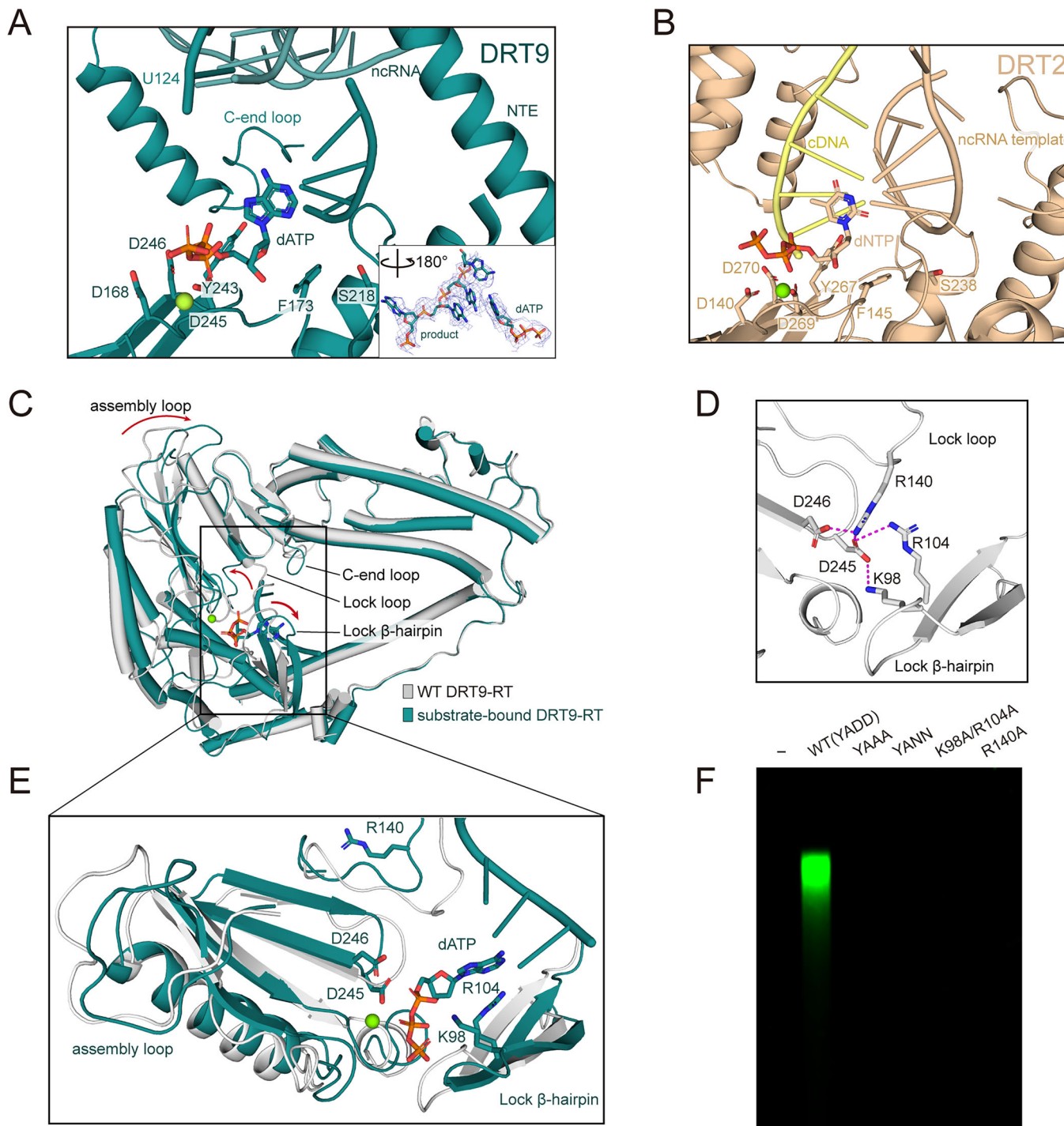

**Figure 5. Structural basis of the enzymatic activity and activation mechanism of the DRT9 system.**

(A, B) Detailed insights into catalytic pocket of DRT9 (deep teal) and DRT2 (wheat). The conserved residues are shown as sticks. (C) Superposition of WT DRT9 RT and dNTP-bound DRT9 RT. The conformational movements are indicated by arrows. (D) Two structural features lock the catalytic residues of DRT9 RT in the substrate-free state. The key interacting residues are shown as sticks. (E) Close-up view of the conformational movements of the DRT9 catalytic pocket. The key interacting residues are shown in stick representation. (F) In vitro reverse transcription assay of DRT9 variants. WT, wild-type DRT9; YAAA and YANN, DRT9 variants with catalytic residues mutated. The gel is representative of three replicates. Source data are available online for this figure.

third dimer binding and driving the assembly transition from a tetrameric to a hexameric state (Figs. 4D and 5C).

In the DRT9 tetramer complex, the catalytic residues are fixed by two structural features, including a lysine-finger lock loop (residues 133–144) and a β-hairpin latch (residues 92-108). Specifically, two basic residues, K98 and R104 on the β-hairpin latch, form hydrogen bonds with D245. The R140 on the lock loop forms hydrogen bonds with D245 and D246 (Fig. 5D). Structural comparison shows that a dATP molecule would sterically clash with these interacting residues in the substrate-free state. Thus, substrate binding induces the displacement of the β-hairpin latch and outward rotation of the lysine-finger lock loop by 180° (Fig. 5E). These coordinated conformational changes disrupt the hydrogen-bonding network, thereby releasing the catalytic residues from their "locked state" and activating DRT9. Strikingly, K98 and R104 on the β-hairpin latch, which "lock" the catalytic aspartates in the substrate-free state, undergo a functional switch upon substrate binding, possibly stabilizing the substrate during catalysis in the dATP-bound state (Fig. 5D,E). K98A/R104A and R140A mutations

impeded cDNA synthesis (Fig. 5F; Appendix Fig. S12A), confirming the significance of these structural features in the "lock-switch" regulation of DRT9 function. Collectively, dNTP binding induces displacement within the catalytic pocket, and allosterically triggers conformational changes in the palm region, especially the loop region for assembly, ultimately leading to higher polymerization and activation of DRT9.

## The C-terminal loop is involved in protein-primed DNA synthesis

In the substrate-free state, the C-terminal loop of RT extends away from the catalytic pocket. Upon dNTP binding, this loop undergoes a ~180° rotation, inserting into a channel adjacent to the four uracil residues (U124–U127) on the ncRNA and the product (Fig. 5A,C). Deletion of the C-terminal loop resulted in the complete loss of DRT9 function, underscoring its essential role in DRT9 activation (Appendix Figs. S11C and S12A). Remarkably, the C-terminal loop

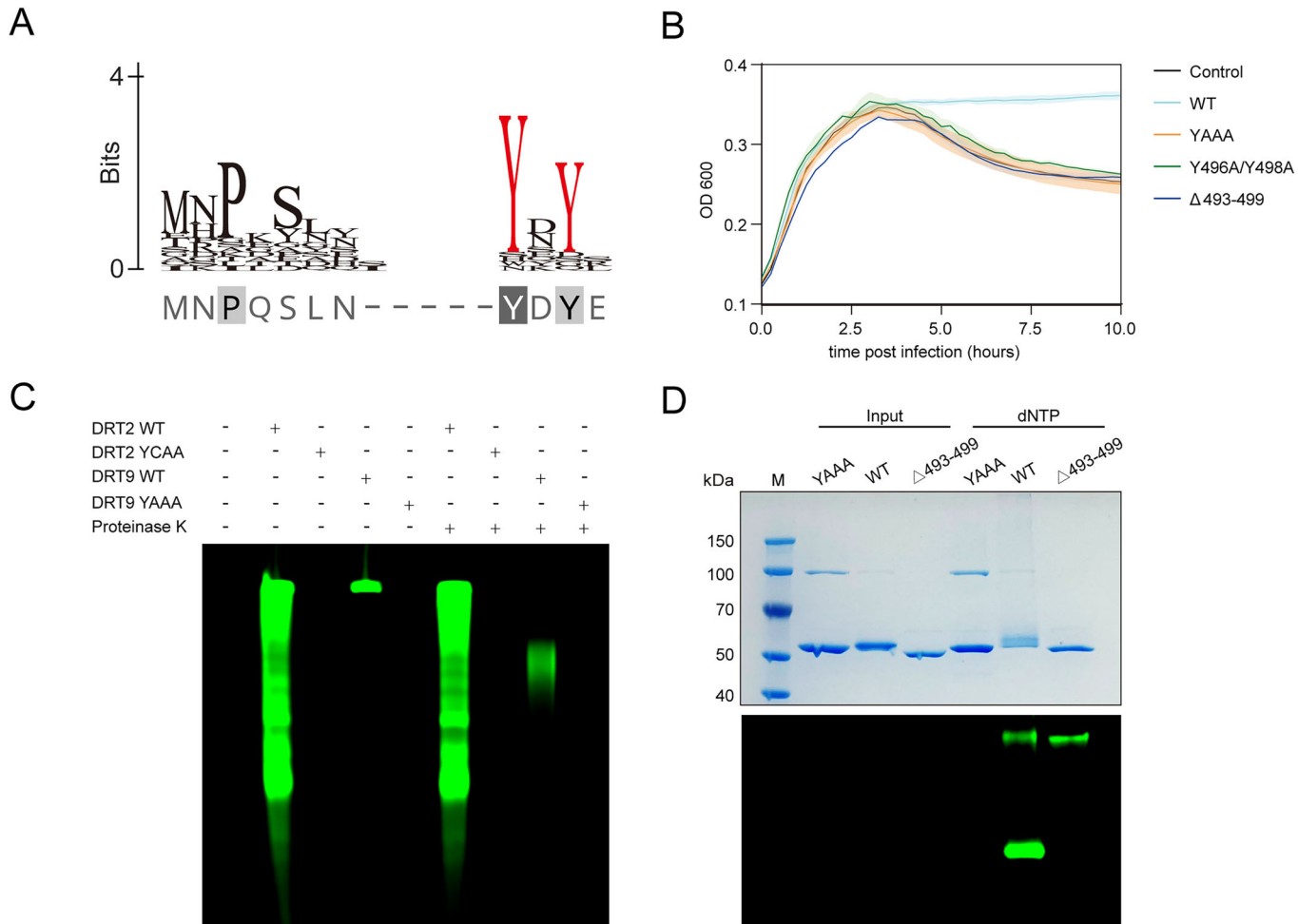

**Figure 6. DRT9 generates cDNA utilizing a different molecular mechanism from DRT2.**

(A) Sequence logo of C-terminus of DRT9 homologs. The conserved residues are shown in red. (B) Growth curves of WT DRT9 and variants. YAAA, catalytically inactive mutant. Y496A/Y498A, DRT9 variant with two conserved Tyr residues mutated. Δ493-499, DRT9 system with C-terminus truncated. (C) In vitro reverse transcription assay of DRT2 and DRT9 variants. Proteinase K is utilized to digest the RT protein. (D) In vitro reverse transcription assay of DRT9 variants. The cDNA products and DRT proteins are visualized with fluorescence imaging and Coomassie Brilliant Blue staining, respectively. All the gels represent three repeat experiments. Source data are available online for this figure.

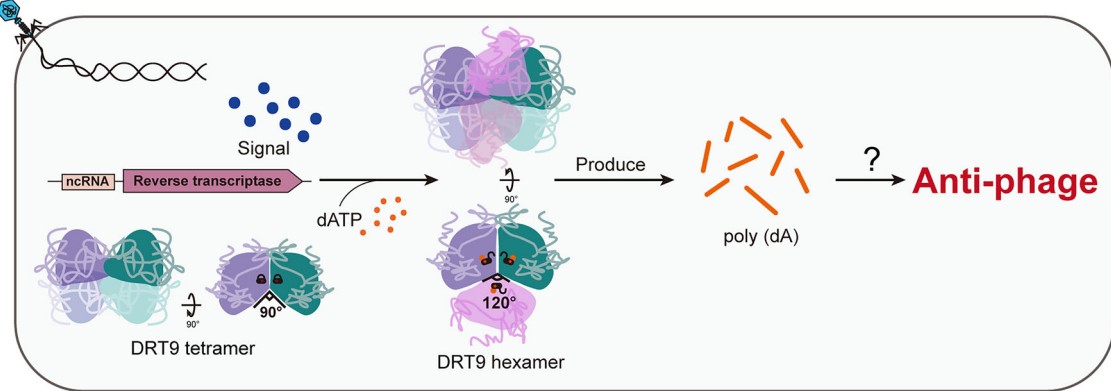

**Figure 7. Schematic diagram illustrating the activation mechanism of the DRT9 system.**

Upon phage infection of cells harboring the DRT9 system, DRT9 tetramers undergo a structural transformation into hexamers, triggered by the binding of dNTPs. These substrate-bound DRT9 hexamers synthesize poly(dA) products, which confer resistance to phage invasion.

of DRT9 contains two tyrosine residues, Y496 and Y498, which are conserved across DRT9 homologs (Fig. 6A). This structural feature resembles AbiK, which initiates reverse transcription through a tyrosine residue-mediated protein-primed mechanism. More importantly, alanine substitution of Y496 and Y498 abolished DRT9-mediated cDNA synthesis and phage defense (Fig. 6B; Appendix Fig. S12B). To further confirm if the reverse transcription of DRT9 is initiated in a protein-primed mechanism, we performed an in vitro reverse transcription assay with proteinase K treatment (Fig. 6C). Untreated DRT9 samples showed no cDNA migration, whereas treated samples displayed distinct product bands (Fig. 6C). By contrast, DRT2 products remained unchanged after proteinase K treatment, as assessed by urea-PAGE. SDS-PAGE analysis further revealed that most DRT9 cDNA co-migrated with the protein band, and this co-migration was abolished upon deletion of the C-terminal loop (Fig. 6D). Taken together, these results demonstrate that DRT9 initiates protein-primed DNA synthesis via its C-terminal loop, with Y496 and Y498 being critical for this process.

## Discussion

The prokaryotic RTs derived from group II intron are believed to provide resistance against phage infection. Although some prokaryotic RTs like Abi RTs autonomously synthesize DNA without other components (Figiel et al, 2022), non-coding RNA (ncRNA) is often required for the enzymatic activity of other RTs, such as the retron, DRT2, and DRT9 systems (Fig. 1A,B). Notably, DRT2 and DRT9 are able to exert anti-phage defense solely through their reverse transcription activity, distinguishing them from other reported ncRNA-facilitated prokaryotic RTs, such as retron systems, which require additional effector modules to mediate anti-phage function (Li et al, 2025). DRT2 system was reported to produce long concatemeric cDNA using a portion of its ncRNA as a template, which is then translated into a toxic protein for phage resistance (Tang et al, 2024; Wilkinson et al, 2024). In contrast, DRT9 specifically generates poly(dA) sequences using dATP as the substrate (Fig. 1E,F).

RT proteins in Abi systems typically function in oligomeric forms for function, AbiA as a dimer, AbiK as a hexamer, and AbiP2 as a trimer. The assembly of these systems generally relies on their helical or HEPN domains. DRT2, by contrast, functions as a monomer. Distinct from these, the assembly of DRT9 is cooperatively mediated by both the RT protein and its associated ncRNA (Fig. 3), and the activation requires a substrate-induced transition from a tetramer (dimer-of-dimers) to a hexamer (trimer-of-dimers) (Fig. 4). Notably, during cryo-EM data processing of DRT9 in the absence of dNTPs, most particles predominantly correspond to tetramers, with only a small fraction (13%) representing hexamers (Appendix Fig. S2). Incubation with dNTPs leads to a significant increase in hexameric DRT9 particles, suggesting a substrate-induced dynamic equilibrium between the tetrameric and hexameric states of DRT9 (Appendix Fig. S8). This transition was further supported by native PAGE, which showed a progressive shift from tetramer to hexamer with increasing dNTP concentrations (Appendix Fig. S7). Interestingly, bacterial dNTP pool concentrations typically range from 20 to 200 μM (Bennett et al, 2009; Buckstein et al, 2008; Wheeler et al, 2005), which are likely to be insufficient to induce the overall complex to transition into the catalytically active hexameric state (Appendix Fig. S7). Therefore, it is plausible that DRT9 remains in an inactive tetrameric state under physiological dNTP level and becomes activated when the dNTP concentration exceeds a critical threshold. The phage-derived factors or secondary signaling molecules could be linked to dNTP-mediated activation (Fig. 7). One possible scenario is that phages encode proteins that inhibit bacterial dNTPase, thereby increasing dNTP levels to facilitate their own replication (Center et al, 1970; Klemm et al, 2022). Alternatively, it has been documented that phage proteins also hijack key components for dNTP synthesis to promote dNTP production (Shen et al, 2004).

Both DRT2 and DRT9 are proposed to defend against phage through cDNA synthesis, but they operate differently. In the DRT2 system, the 3' end of ncRNA is inserted into the catalytic center of the palm part for priming (Wilkinson et al, 2024) (Fig. 5B). By contrast, the fingers region of RT holds the 3' side of ncRNA in

DRT9 system (Fig. 5A; Appendix Fig. S5). Two structural elements, including a hinge-like loop and a β-hairpin latch, regulate DRT9 function through a dynamic "lock-switch" mechanism (Fig. 5). In the substrate-bound state of DRT9, cryo-EM density indicates that the C-terminal loop may occupy the same position as the cDNA primer in DRT2, raising the possibility that this C-terminal loop is implicated in priming, reminiscent of the protein (tyrosine and serine)-priming observed in Abi RTs (Figiel et al, 2022). Indeed, multiple conserved tyrosine residues are found in this C-terminal loop. More importantly, our SDS-PAGE results show that most cDNA products of DRT9 coincide with the RT protein band, but deleting the C-terminal loop eliminates this product, which contrasts with the results in DRT2 system, demonstrating the protein-primed cDNA synthesis of DRT9 (Fig. 6). However, we did not observe cryo-EM density corresponding to a potential covalent linkage between the C-terminal residues of DRT9 and the product. It is likely due to the intrinsic flexibility of the C-terminal region, which may facilitate cDNA elongation—a mechanism that has been proposed for Abi RTs (Figiel et al, 2022; Gapinska et al, 2024). Moreover, further investigation is needed to elucidate how the poly(dA) products trigger downstream bacterial immune responses to counter phage infection. Taken together, our work elucidates the biochemical and structural mechanisms underlying DRT9-mediated anti-phage defense, expanding the understanding of reverse transcriptases in bacterial immunity.

# Methods

### Reagents and tools table

| Reagent/resource | Reference or source | Identifier or catalog number |
|---|---|---|
| **Experimental models** | | |
| *Escherichia coli* strain DH5α | Tsingke Biotechnology | TSC-C14 |
| *Escherichia coli* BL21 (DE3) | Tsingke Biotechnology | TSC-E01 |
| **Recombinant DNA** | | |
| 2CT-10_DRT9 vector | This study | |
| pET28-MKH8SUMO_DRT2 vector | This study | |
| 13S-A_ncRNA vector | This study | |
| **Chemicals, enzymes and other reagents** | | |
| HiTrap Q HP column | Cytiva | 17115401 |
| Superose™ 6 Increase 10/300 GL | Cytiva | 29091596 |
| **Software** | | |
| Cryosparc | Punjani et al (2017) | |
| AlphaFold3 | Abramson et al, (2024) | |
| Phenix | Williams et al (2018) | |
| Coot | Emsley et al, (2010) | |
| PyMOL | https://pymol.org/ | |
| UCSF ChimeraX | Meng et al, (2023) | |
| MAFFT | Katoh et al (2002) | |

| Reagent/resource | Reference or source | Identifier or catalog number |
|---|---|---|
| trimAl | Capella-Gutierrez et al (2009) | |
| IQTREE | Minh et al (2020) | |
| iTOL | Letunic and Bork, (2024) | |
| Bowtie2 | Langmead et al (2019) | |
| seqtk | https://github.com/lh3/seqtk | |
| MEME Suite | Bailey et al (2015) | |
| **Other** | | |
| ÄKTA pure | Cytiva | N/A |
| Visible Fluorescent Imager | Azure Biosystems | N/A |

## Protein expression and purification

The *drt9* gene encoding full-length DRT9 protein (GenBank: EAO1508900.1) was amplified by PCR and cloned into the 2CT-10 vector (Addgene #55209) with an N-terminal His10-MBP fusion tag. The *drt2* gene encoding the full-length DRT2 protein (GenBank: NC_012731.1) was cloned into the pET28-MKH8SUMO vector (Addgene #79526) with an N-terminal His8-SUMO fusion tag. The ncRNA sequences of DRT9 and DRT2 were cloned into a 13S-A vector (Addgene #48323), respectively. For expression and purification of DRT9, the prepared vectors, which contain components of DRT9 system, were co-transformed into BL21 (DE3) cells with 0.5 mM IPTG added to induce protein expression. When the $OD_{600}$ value reached 0.6, the cells were harvested through centrifugation and lysed by sonication in lysis buffer (25 mM Tris-HCl pH 7.5, 500 mM NaCl, 2 mM β-mercaptoethanol). The supernatant of lysate was loaded onto Ni-NTA resin (Qiagen) and the target protein was eluted by elution buffer (25 mM Tris-HCl pH 7.5, 500 mM NaCl, 2 mM β-mercaptoethanol, 300 mM imidazole). TEV was utilized to remove the fusion tag. The target protein complex was further purified with HiTrap Q HP column (Cytiva) for ion-exchange chromatography and Superose 6 Increase column (Cytiva) with gel-filtration buffer (25 mM Tris-HCl pH 7.5, 150 mM NaCl, 2 mM DTT). The peak fractions containing target protein were collected and further analyzed with SDS-PAGE and urea-PAGE. The expression and purification procedures for the DRT2 system are similar to those used for DRT9.

## Phylogenetic analysis

To investigate the phylogenetic relationships of the DRT family related to anti-phage defense, we constructed a protein sequence dataset comprising DRT family members and Abi family proteins with experimentally validated anti-phage activity for analysis (Mestre et al, 2022). Multiple sequence alignment was performed using MAFFT v7.526 with default parameters (Katoh et al, 2002). The results were trimmed to remove excessive gap regions using trimAl v1.5 (-gappyout) (Capella-Gutierrez et al, 2009). Phylogenetic tree was constructed using IQTREE v2.3.6 (Minh et al, 2020),

with the Modelfinder module selecting Q.pfam+F + I + R10 as the optimal model (Kalyaanamoorthy et al, 2017). Branch support was assessed through 1000 iterations of ultraFast bootstrap (UFBoot) resampling (Hoang et al, 2018). The tree was visualized using the iTOL v7 platform (Letunic and Bork, 2024).

## Sequencing of cDNA products

Raw sequencing fragments were aligned to a custom reference genome (DRT9_ncRNA) using Bowtie2 (v2.5.1) (Langmead et al, 2019). Unmapped reads were extracted and further analyzed. The nucleotide composition of unmapped reads was first assessed using seqtk comp (https://github.com/lh3/seqtk), followed by removal of poly-C tails, which may arise from reverse transcriptase terminal transferase activity. The trimmed reads were re-evaluated for base composition to determine correction efficiency. To quantify ambiguous base ($N$) frequencies, we applied a Python-based approach that iterated through each sequence to count the number of '$N$' bases per read and normalized the values as counts per million (CPM) based on total read numbers. Additional analysis of sequence bias was performed using custom Python scripts to evaluate both homopolymer prevalence and nucleotide positional distribution. For homopolymer detection, each read was scanned for stretches of consecutive identical nucleotides (A, T, C, or G) with lengths ≥5, 10, 15, 20, 25, 30, 35, and 40 nt. The frequency of each motif type was calculated as a percentage of the total read count. To assess position-specific nucleotide bias, we analyzed base composition at each position (up to 122 nt) across all reads. Each nucleotide (A, T, C, G, N) at every position was counted, and its frequency was computed by normalizing against the total number of bases at that position, yielding a per-position nucleotide frequency profile. Finally, unmapped reads were converted to FASTA format, filtered for poly(A)-containing sequences, and clustered by sequence identity. Sequences with at least ten identical copies were retained and used for de novo motif discovery using MEME Suite (v5.5.0) (Bailey et al, 2015) under the ZOOPS model with a motif width of 35–40 bp. Motif logos were generated using meme2images.

## Multiple sequence alignment of DRT9 ncRNA sequences

Based on the predicted ncRNA locations that have been reported and the ncRNA length of DRT9 system, 300 bp nucleotide sequences upstream of DRT9 orthologs were extracted and aligned with DRT9 ncRNA using MAFFT (Katoh and Standley, 2013). The gaps of DRT9 ncRNA in the multiple sequence alignment results were removed by trimAl. The analysis of processed results and the logo plot were achieved by Jalview (Waterhouse et al, 2009).

## In vitro reverse transcription assay

The purified protein complex at a final concentration of 3 μM was added with 100 μM dNTP mixed with 5 μM fluorescein-12-dATP (AAT Bioquest). The reaction buffer (25 mM Tris-HCl pH 7.5, 150 mM NaCl, 5 mM MgCl₂, 5 mM DTT) was supplemented to 20 μl. The reaction occurred at 37 °C for 1 h and was terminated by proteinase K. After 15 min, proteinase K was heated to be denatured so that the RNA was digested by 0.2 μl RNase A at 37 °C for 15 min. The processed sample was further analyzed by

12% urea-PAGE gel. The fluorescence signal was analyzed using an Azure Imaging System.

## Native polyacrylamide gel electrophoresis (PAGE)

The target protein complex was incubated with dNTP at a molar ratio of 3:1000 at room temperature for 1 h. The sample was separated by 4% native PAGE and analyzed for aggregation state by Coomassie Brilliant Blue staining.

## Protein-nucleic acid covalent binding assay

To visualize the covalent bond of nucleic acids to DRT9 RT, DRT9 complex was incubated with 100 μM dNTPs mixed with 5 μM Fluorescein-12-dATP in a reaction buffer described above for 1 h at 37 °C. The reaction was terminated with 0.2 μl RNase A. Protein samples were denatured by heating at 98 °C for 3 min and subsequently analyzed on a 12% SDS-PAGE gel. The gel was then scanned for fluorescence signals and stained with Coomassie Brilliant Blue.

## Phage amplification and plaque assays

MG1655 cells were cultured in LB medium at 37 °C, 200 rpm for 16 h supplemented with 1 mM CaCl₂ and 1 mM MgCl₂. The overnight cultures were diluted 1:100 with fresh LB medium and incubated at 37 °C until the OD₆₀₀ reached 0.4. T5 phage was then added at an MOI of 0.1, and the culture was harvested when it became clear. The T5 phage was collected by centrifugation at 13,000 rpm for 25 min, followed by filtration utilizing a 0.22 μm filter, before stored at 4 °C.

DRT9 and ncRNA were individually cloned into the pBAD-LIC (Addgene #37501) and 13S-A (Addgene #48323) vectors, respectively. Both plasmids were then transformed into *Escherichia coli* BL21-AI. Overnight cultures of the transformed bacteria were diluted 1:100 in LB medium and incubated at 37 °C, 200 rpm until the OD₆₀₀ reached 0.6. The cultures were then induced with 0.2% arabinose and 1 mM IPTG for 4 h. Subsequently, 200 μl of the induced culture was mixed with 10 ml of pre-melted semi-solid agar medium containing 1 mM CaCl₂, 1 mM MgCl₂, 0.2% arabinose, and 1 mM IPTG, and poured onto the surface of a 10 cm Petri dish pre-coated with solid agar. T5 phage was serially diluted 10-fold in PBS and added to the top agar layer. The plates were then inverted and incubated overnight at 37 °C.

## Growth assays

Overnight cultures of BL21-AI were diluted and induced with 0.2% arabinose and 1 mM IPTG until the OD₆₀₀ reached 0.2. 180 μl of culture was mixed with 20 μl of T5 phage dilutions in LB to achieve final MOIs of 1, 0.1, and 0.01, then transferred to 96-well plates and incubated at 37 °C with shaking. The OD₆₀₀ was measured every 15 min for 10 h using a microplate reader (Biotek).

## Cryo-EM sample preparation and imaging

For DRT9 tetramer sample and DRT2 system sample, an aliquot of 3.5 μl purified target proteins at 2 mg/ml was applied to glow-discharged Cu R1.2/1.3 holey carbon grid (200 mesh, Quantifoil).

After a 20 s incubation, grids were blotted with a force of 0 for 2 s at 4 °C and 100% humidity, and plunge-frozen into liquid ethane using Vitrobot Mark IV (FEI Thermo Fisher). To obtain the structure of substrate-bound DRT9 complex, 1 mM of each dNTPs was added to DRT9 (2 mg/ml) in a buffer containing 25 mM Tris-HCl pH 7.5, 150 mM NaCl, and 5 mM MgCl$_2$. The mixture was incubated at room temperature for 1 h, after which 3.5 μl of the mixture was applied for cryo-EM sample preparation using the same procedure.

All grids were imaged using a CRYO ARM 300 electron microscope (JEOL, Japan) operating at 300 kV equipped with a K3 direct electron detector (Gatan, USA). Cryo-EM images were acquired automatically using Serial-EM software (Mastronarde, 2005) with a super-resolution pixel size of 0.475 Å/pixel at defocus values ranging from −0.5 to −2.5 μm at a calibrated magnification of ×50,000. Data were collected at a frame rate of 40 frames per second. The total electron dose was 40 e$^-$/Å$^2$.

### Cryo-EM data processing

All cryo-EM data were processed with cryoSPARC (Punjani et al, 2017). Recorded movies were subjected to patch motion correction and followed by contrast transfer function (CTF) estimation.

For DRT9 tetramer complex, 6625 micrographs were collected and imported to CryoSPARC. Particles were picked using blob picking and template picking successively, followed by multiple rounds of 2D classification. 222,152 particles were selected to train a Topaz model for particle picking (Bepler et al, 2019). Meanwhile, these particles were used to generate the initial volumes. Using this Topaz model, particles were picked from 6625 micrographs, and good particles were selected through 2D classification, yielding 241,593 particles. After combining the particle stacks from three picking methods, heterogeneous refinement and non-uniform (NU) refinement were utilized to improve the quality of initial volumes, generating the final map of DRT9 tetramer complex at 3.49 Å.

For the substrate-bound DRT9 complex, 7850 micrographs were collected and particles were picked independently using blob picking, template picking and Topaz picking. A final dataset of 10,292,180 particles was generated and multiple rounds of 2D classification were carried out, yielding 906,764 good particles for ab initio reconstruction. Two volumes featuring obvious secondary structures were used as initial volumes for following heterogeneous refinement. Subsequent rounds of 3D classification were conducted to isolate distinct hexameric and tetrameric states. After NU-refinement, the substrate-bound DRT9 hexamer (40,373 particles) generated a cryo-EM map at 3.46 Å. We also identified a population of hexamer particles lacking clear dATP and product densities. Therefore, we did not pursue additional data processing.

For the DRT2 system, 91,610 particles were picked from 213 micrographs using blob picking, extracted with the same box size (256 pixels) as applied in DRT9 processing, and subjected to classification to generate 2D averages.

### Model building and refinement

The initial model of DRT9 was acquired from Alphafold3 (Abramson et al, 2024) provided with the sequences of DRT9 and ncRNA. Then tracing the density map, the protein model was manually refined to fit the density map and the ncRNA was manually mutated and refined by Coot (Emsley et al, 2010) to map

the sequence. Iterative refinements were executed with Coot and phenix.real_space_refine (Afonine et al, 2018) to generate the final model. For the model building of substrate-bound DRT9 complex, six monomers split from the hexamer were docked into the volume map separately and combined in ChimeraX (Meng et al, 2023). The combined model was manually refined by Coot and phenix.real_-space_refine was utilized for further refinement. The quality of all models was validated with MolProbity (Williams et al, 2018) and concluded in Appendix Table S1.

## Data availability

Next-generation sequencing data have been deposited in NCBI (BioProject: PRJNA1280644). The atomic coordinates have been deposited in the Protein Data Bank (https://www.rcsb.org/) under accession codes 9VKU (DRT9 tetramer complex) and 9VMA (substrate-bound DRT9 complex). Cryo-EM maps have been deposited in the Electron Microscopy Data Bank (https://www.ebi.ac.uk/emdb/) under corresponding accession codes EMD-65143 and EMD-65181. Custom scripts used for bioinformatics are available upon request.

The source data of this paper are collected in the following database record: biostudies:S-SCDT-10_1038-S44318-025-00544-8.

## Peer review information

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

## Acknowledgements

We thank the Center for Instrumental Analysis and Metrology of Wuhan Institute of Virology for supporting cryo-EM data acquisition and the Core Facility of Research Center of Basic Medical Sciences in Tianjin Medical University for providing technical assistance. This work was supported by the National Natural Science Foundation of China (32322040 to HZ and 32300036 to HY), Tianjin Science and Technology Plan Project (23ZYCGSY00750), the Natural Science Foundation of Tianjin Municipal Science and Technology Commission (23JCZDJC00410), Scientific Research Program of Tianjin Municipal Education Commission (2023ZD011), the CAS Pioneer Hundred Talents Program (to ZD), and Youth Research Incubation Fund of School of Basic Medical Sciences, Tianjin Medical University (2023FY07).

## Author contributions

**Jie Han**: Funding acquisition; Investigation; Methodology. **Bin Liu**: Investigation; Methodology. **Jingjing Tang**: Data curation; Software; Investigation; Visualization. **Shuqin Zhang**: Software; Visualization; Writing—original draft. **Xiaoshen Wang**: Data curation; Validation; Investigation; Visualization; Methodology. **Xuzichao Li**: Data curation; Software; Validation; Visualization; Writing—original draft. **Qian Zhang**: Investigation; Methodology. **Zhikun Liu**: Software; Visualization; Writing—original draft. **Wanyao Wang**: Data curation. **Yingcan Liu**: Software; Investigation; Methodology. **Ruimin Zhou**: Data curation. **Hang Yin**: Resources; Formal analysis; Supervision. **Yong Wei**: Resources; Data curation. **Zhuang Li**: Formal analysis; Investigation; Methodology. **Minjie Zhang**: Supervision; Investigation; Visualization; Methodology. **Zengqin Deng**: Resources; Data curation; Formal analysis; Supervision; Funding acquisition; Validation; Visualization; Project administration. **Heng Zhang**: Conceptualization; Resources; Data curation; Supervision; Funding acquisition; Project administration; Writing—review and editing.

Source data underlying figure panels in this paper may have individual authorship assigned. Where available, figure panel/source data authorship is listed in the following database record: biostudies:S-SCDT-10_1038-S44318-025-00544-8.

## Disclosure and competing interests statement

The authors declare no competing interests.

