## [Peer Review File · The EMBO Journal]

Non-coding RNA mediates the Defense-Associated Reverse Transcriptase (DRT) anti-phage oligomerization transition

Jie Han, bin liu, jingjing tang, shuqin zhang, Xiaoshen Wang, xuzichao li, qian zhang, zhikun liu, wanyao wang, Yingcan Liu, Ruimin Zhou, Hang Yin, yong wei, zhuang li, Minjie Zhang, Zeng Deng, and Heng Zhang

Corresponding author(s): Heng Zhang (zhangheng134@gmail.com) , Zeng Deng (dengzengqin@wh.iov.cn), Minjie Zhang (zhangmj@tmu.edu.cn)

Review Timeline:

Submission Date:	26th Jan 25
Editorial Decision:	22nd Apr 25
Revision Received:	16th Jul 25
Accepted:	6th Aug 25

Editor: Hartmut Vodermaier

Transaction Report:

Dear Heng,

Since we have still not received comments from the third referee on your DRT9 manuscript, I decided to already forward you the reports of the other two reviewers now, copied below. Both of them consider the topic of your study very interesting and especially your structural data potentially important. Nevertheless, they share overlapping concerns about the experimental support for the proposed overall model, in particular on the physiological significance of particular oligomeric states of DRT9. These aspects would in my opinion need to be decisively addressed in order to justify publication in The EMBO Journal; while without this, the structural/functional characterization in the present form and with only minor adjustments may be better suited for our sister journal EMBO Reports.

Since it is not clear if and how the main issues could be satisfactorily addressed during a regular revision, I would at this point invite you to go through the two reports and to then send me a tentative point-by-point response, detailing how you would envision clarifying the key concerns of the referees. Based on such a revision proposal, we could then discuss whether a major revision for The EMBO Journal would seem realistic, or whether the study would be better candidate for rapid publication in EMBO Reports. Obviously, this final decision may also be influenced by any comments from the outstanding 3rd reviewer, in case we should still receive them in the coming days.

Once more apologies for the substantial delay with this external review, and I look forward to hearing from you within the next two weeks,

Best regards,

Hartmut

Referee #1 (Report for Author)

In the manuscript titled "Non-coding RNA mediates the Defense-Associated Reverse Transcriptase anti-phage oligomerization transition," the authors systematically and functionally characterized an *Escherichia coli* DRT9 system. They demonstrated that both the reverse transcriptase and non-coding RNA are crucial for the anti-phage function. To investigate its biochemical activity, the authors showed that DRT9 is involved in cDNA synthesis, which is hypothesized to form a covalent bond through the utilization of the C-terminal tail of the reverse transcriptase protein. Furthermore, using cryogenic electron microscopy, the authors determined two cryoEM structures of the DRT9 complex, one with and one without the substrate. Finally, the authors proposed a regulatory mechanism. They suggested that upon substrate binding, the DRT9 complex undergoes a transition, where the dimer-of-dimer complex changes to a trimer-of-dimer complex.

The authors present a comprehensive structural analysis of DRT9 complexes, examining two distinct forms. They delve into the structural changes and propose an intriguing model that suggests how the DRT9 system facilitates the anti-phage function. While the structural and functional characterization is meticulously detailed, the proposed model lacks compelling experimental evidence and a clear rationale for considering the dimer-of-dimer state as physiologically relevant. Below are my major concerns and suggestions aimed at enhancing the quality of this manuscript, and I hope the authors find them helpful.

Major points:

The authors determined the sequence of cDNA from their *in vitro* reconstituted reverse transcription reactions. However, they did not discuss the mapping of these sequence reads in this manuscript. Do these map to non-coding RNA (ncRNA) sequence? If not, are they similar to ncRNA-expressing plasmid regions? Understanding what the synthesized cDNA codes for (if it does) is crucial to understanding the mechanism of DRT9. At the very least, the authors should submit and share their sequencing data in a publicly accessible repository and provide an explanation for their inability to map these reads.

Could the authors please clarify how the size of the DRT9 polymer ("~450 kDa") was determined in Figure S1b? Were protein standards used for this calculation, or is it based

on multi-angle light scattering or other techniques?

The "Material and Methods" section is missing crucial information about how the DRT2 was expressed, purified, and the CryoEM data collection for 2D classes in Figure 1d. The author should either provide this information or cite the appropriate sources.

To understand the functional importance of tetrameric assembly, the authors aimed at disrupting the oligomer using ncRNA mutagenesis (G114C) and by deleting the assembly loop region 288-299. These alterations were analyzed using native PAGE analysis, but the results don't support the claim stated "Deletion of the assembly loop (D288-299) disrupts the higher-order oligomerization of DRT9." For instance, the wild-type complex, which forms a tetramer in the absence of dNTP, exhibited similar gel shift patterns to the two mutants studied. Could the author clarify why these mutants and the wild-type complex move at the same size? Alternatively, could these discrepancies be resolved by employing a size exclusion-based assay, as demonstrated for the wild-type complex in Figure S1b. The current characterization of these mutants does not support the disruption of tetrameric assembly.

The authors employed D3 symmetry during the cryoEM data processing of the substrate-bound DRT9 complex. Could they have observed the cDNA density in all the protomers before processing as a D3 complex? Furthermore, did the authors attempt performing symmetry expansion to enhance the structure resolution further? Could the authors show the additional cryoEM density they mentioned for the cDNA product? Perhaps symmetry expansion could aid in resolving the cDNA density to a higher resolution. Lastly, it remains unclear why a significant portion of particles (77.8%) were excluded from further data processing in Figure S7. An explanation for this exclusion should be provided.

The authors showed a dimer-of-dimer to trimer-of-dimer transitions in DRT9 in the presence of dNTP mix. The physiological role or existence of the dimer-of-dimer state is unclear to me, as the bacterial dNTP pools would likely maintain the overall complex in a trimer-of-dimer state. The authors should discuss why the tetrameric structure would be physiologically relevant.

DRT9-mediated cDNA synthesis is one of the most exciting findings in this study (Figure S11). If indeed there was no sequence specificity in the cDNA, this could suggest cDNA-mediated regulation in this family of proteins. Could the author please provide the density for the covalent adjunct between the DRT9 C-terminal residues (Ser/Tyr) and cDNA? Additionally, I would suggest that Figure S11 be moved to the main figures as it is one of the

impactful observation in this study. Furthermore, I would insist on performing protein sequence alignment among multiple DRT9 homologs to identify the critical amino acid involved in priming reaction. Analyzing the growth curve of priming crucial mutants might offer functional insights if DRT9 is abortive in nature and if the random cDNA serves as a regulatory mechanism in this scenario.

Please note that in the alignment depicted in Figure S9, there's an extra N-terminal His tag sequence that might be disrupting the alignment. Removing this tag could potentially enhance the alignment features.

The manuscript discussion abruptly ends. It should be resumed to discuss how the current manuscript advances our understanding of DRTs and establishes a solid foundation for a distinct mechanism. It should also address unanswered futuristic questions that arise from this study.

Here are some minor but important points:

Please label the organisms' names in the schematics shown in Figure 1b.

What do the authors mean by "The gel is representative of three biological replicates" (Figure 1c legend). This is an *in vitro* assay, please fix this statement.

Please add pixel size to the legend of Figure 1d.

Please mention the sequence similarity between DRT2 and DRT9 in the text.

Could you please make the PFU assays more visually clear throughout the manuscript? These are hard to see.

There's a typo in Figure 3d and S8a. It should be "assembly".

In Table 1, under the heading "Post-Refinement," what does the author mean by "model resolution"? How was this calculated?

The title of the section "ncRNA conservative analysis" should be corrected.

In the "In vitro reverse transcription assay" section of the Material and Methods, please

provide the percentage of Urea PAGE.

Grammatical errors, superscripts, and other typos need to be corrected throughout the manuscript.

Referee #2 (Report for Author)

In this manuscript, Han et al. present a structural and functional investigation of the DRT9 anti-phage defense system, which consists of a reverse transcriptase (RT) and a non-coding RNA (ncRNA). Using cryo-EM, the authors determined molecular structures of the DRT9 RT-ncRNA complex, which adopts distinct conformational and oligomeric states depending on the presence of the substrate, dNTPs. Based on these structural observations combined with functional assays, the authors propose that the dNTP binding induces a conformational rearrangement and an oligomeric transition, ultimately leading to the exposure of the active site and catalytic activation of the RT in the DRT9 system.

This study is of considerable significance, as it provides novel structural insights into the DRT9 system and proposes a unique activation mechanism supported by the structural and biochemical data. The cryo-EM reconstructions are of high quality and contribute valuable information to the field. However, several aspects of the data interpretation remain somewhat ambiguous, and additional analyses or discussion would significantly enhance the clarity and overall impact of the manuscript.

Major concerns

1. The cryo-EM data clearly show that the DRT9 RT-ncRNA complex forms tetramers in the absence of dNTPs while transitioning into hexamers upon dNTP binding. Based on this observation, the authors propose that this tetramer-to-hexamer transition is essential for enzymatic activation and the anti-phage defense function of DRT9. However, the precise mechanism underlying this transition is not well explained. Does substrate binding cause dissociation of tetramers into dimers, which subsequently assemble into hexamers? Were such intermediate dimers observed upon dNTP addition during the cryo EM or other experiments? Alternatively, is there a pre-existing equilibrium between tetramers and dimers in the absence of dNTPs? The SEC data shown in Figure S1b (without dNTPs) suggest a monodisperse population, which appears consistent with a single, likely tetrameric state. The authors are encouraged to explore these possibilities and, if necessary, provide additional data or discussion to clarify the mechanistic details of this oligomeric transition.

2. In addition to the cryo EM analysis, the authors performed biochemical experiments, in which the oligomeric state of the DRT system could be estimated. It would be helpful to clarify whether the oligomeric states observed in these biochemical assays are fully consistent with the cryo-EM results. For example, in Figure S1b, the authors indicate a peak corresponding to "DRT9 polymer ~450 kDa" but do not explicitly assign this to a specific oligomeric state. Similarly, in Figure S6b, the band shifts observed upon dNTP addition are simply labeled as "DRT9" and "DRT9 oligomer," without clear identification of their stoichiometry. This information would strengthen the mechanistic interpretation. Additionally, the authors should consider investigating whether the loss-of-function mutants, examined throughout the manuscript, show defects in oligomeric assembly. Such analyses would provide further insight into the functional relevance of the mutated residues.

3. It would also be helpful to include a quantitative discussion on the dNTP concentration required for DRT9 activation, considering that dNTPs are constitutively present in bacterial cells. Is there a threshold dNTP concentration necessary to trigger activation? In Figure 6, the authors refer to a "Signal", presumably representing phage-derived factors or secondary signaling molecules that initiate the anti-phage response. Could this "signal" be linked to dNTP-mediated activation? Discussing the potential interplay between these factors would strengthen the biological relevance of the proposed mechanism.

Minor points

1. (Page 5, Lines 16-17) The authors describe that "DRT2 generates long cDNA product, typically larger than 450 nt. If this observation comes from a previous study, a proper reference should be provided. Additionally, it would be helpful to include a DNA size ladder in Figure 1C for clarity.

2. (Page 6, Line 2) There appears to be inconsistency in the length of the ncRNA in the DRT9 system-listed as 188 nt in the text, but shown as 180 nt in Figure 1b. This discrepancy should be clarified.

3. (Page 9, Line 24) It would enhance the clarity of the structural data if the cryo-EM densities for the bound dNTPs and cDNA products were shown explicitly in a figure panel.

Response to Reviewer Comments

We thank the reviewers for their positive and insightful comments, which are invaluable in improving our manuscript. Our proposed responses to the specific points are detailed below. We hope our planned experiments can address the reviewers' concern.

Referee #1 (Report for Author)

In the manuscript titled "Non-coding RNA mediates the Defense-Associated Reverse Transcriptase anti-phage oligomerization transition," the authors systematically and functionally characterized an *Escherichia coli* DRT9 system. They demonstrated that both the reverse transcriptase and non-coding RNA are crucial for the anti-phage function. To investigate its biochemical activity, the authors showed that DRT9 is involved in cDNA synthesis, which is hypothesized to form a covalent bond through the utilization of the C-terminal tail of the reverse transcriptase protein. Furthermore, using cryogenic electron microscopy, the authors determined two cryoEM structures of the DRT9 complex, one with and one without the substrate. Finally, the authors proposed a regulatory mechanism. They suggested that upon substrate binding, the DRT9 complex undergoes a transition, where the dimer-of-dimer complex changes to a trimer-of-dimer complex.

The authors present a comprehensive structural analysis of DRT9 complexes, examining two distinct forms. They delve into the structural changes and propose an intriguing model that suggests how the DRT9 system facilitates the anti-phage function. While the structural and functional characterization is meticulously detailed, the proposed model lacks compelling experimental evidence and a clear rationale for considering the dimer-of-dimer state as physiologically relevant. Below are my major concerns and suggestions aimed at enhancing the quality of this manuscript, and I hope the authors find them helpful.

RE: We sincerely appreciate the reviewer's valuable feedback and have carefully addressed each comment below.

Major points:

1. The authors determined the sequence of cDNA from their in vitro reconstituted reverse transcription reactions. However, they did not discuss the mapping of these sequence reads in this manuscript. Do these map to non-coding RNA (ncRNA)

sequence? If not, are they similar to ncRNA-expressing plasmid regions? Understanding what the synthesized cDNA codes for (if it does) is crucial to understanding the mechanism of DRT9. At the very least, the authors should submit and share their sequencing data in a publicly accessible repository and provide an explanation for their inability to map these reads.

RE: We appreciate the reviewer's insightful comment. During the peer review process, we made additional efforts to characterize the synthesized cDNA codes. Although the sequence of the synthesized DNA appears random, we found that only dATP, rather than the other three dNTPs, is necessary for DRT9 to synthesize DNA (**Fig. R1**). Indeed, we observed a high abundance of polyA sequences (~150,000 reads) in our sequencing data. A comprehensive analysis of the sequencing read mapping will be included in the revised version.

Fig. R1 In vitro reverse transcription assay of DRT9 in the presence of individual dNTPs.

2. Could the authors please clarify how the size of the DRT9 polymer ("~450 kDa") was determined in Figure S1b? Were protein standards used for this calculation, or is it based on multi-angle light scattering or other techniques?

RE: We apologize for the confusion. The molecular weight of the DRT9 polymer was estimated using size-exclusion chromatography (SEC) with protein standards. We are performing analytical ultracentrifugation (AUC) or SEC-MALS to further confirm the oligomeric state of DRT9.

3. The "Material and Methods" section is missing crucial information about how the DRT2 was expressed, purified, and the CryoEM data collection for 2D classes in Figure 1d. The author should either provide this information or cite the appropriate sources.

RE: Thanks for pointing this out. We will add these information in "Material and Methods" section in the revised manuscript.

4. To understand the functional importance of tetrameric assembly, the authors aimed at disrupting the oligomer using ncRNA mutagenesis (G114C) and by region 288-299. These alterations were analyzed using native PAGE analysis, but the results don't support the claim stated "Deletion of the assembly loop (D288-299) disrupts the higher-order oligomerization of DRT9." For instance, the wild-type complex, which forms a tetramer in the absence of dNTP, exhibited similar gel shift patterns to the two mutants studied. Could the author clarify why these mutants and the wild-type complex move at the same size? Alternatively, could these discrepancies be resolved by employing a size exclusion-based assay, as demonstrated for the wild-type complex in Figure S1b. The current characterization of these mutants does not support the disruption of tetrameric assembly.

RE: We apologize for the confusion. The intended information was not clearly stated in our original version, leading to a misunderstanding. The 94C–114G Watson-Crick base pairs and the assembly loop region (residues 288–299) are critical for the structural transition from tetramer to hexamer. However, mutations in these elements do not alter tetramer assembly. To avoid misunderstanding, we will revise the statement "Deletion of the assembly loop disrupts the higher-order oligomerization of DRT9" to "Deletion of the assembly loop disrupts the tetramer-to-hexamer transition of DRT9." The descriptions are imprecise and will be modified accordingly in the revised version. Additionally, the oligomerization status of these mutations will be investigated using AUC and SEC-MALS in our revision.

5. The authors employed D3 symmetry during the cryoEM data processing of the substrate-bound DRT9 complex. Could they have observed the cDNA density in all the protomers before processing as a D3 complex? Furthermore, did the authors attempt performing symmetry expansion to enhance the structure resolution further? Could the authors show the additional cryoEM density they mentioned for the? Perhaps symmetry expansion could aid in resolving the cDNA density to a higher resolution. Lastly, it remains unclear why a significant portion of particles (77.8%) were excluded from further data processing in Figure S7. An explanation for this exclusion should be provided.

RE: As suggested by the reviewer, we re-processed the cryo-EM data without imposing symmetry (C1 symmetry) (**Fig. R3**) and confirmed the presence of cDNA density in the substrate-bound complex (**Fig. R4**), which is consistent with our previous model. However, the resolution obtained with symmetry expansion (3.46 Å) is lower than that obtained with D3 symmetry (3.0 Å), therefore, the previous map is used for analysis.

Fig. R3 Cryo-EM data processing of the DRT9-dNTP complex.

Fig. R4 Cryo-EM density of cDNA.

As mentioned by the reviewer, unambiguous dNTP/cDNA density are traced in the

22.1% of hexamer particles corresponding to the substrate-bound structure, while dNTP/cDNA are absent in the remaining (77.8%) hexamer particles, which may represent a pre-synthesis state poised for catalysis. This analysis will be detailed in the revised manuscript.

6. The authors showed a dimer-of-dimer to trimer-of-dimer transitions in DRT9 in the presence of dNTP mix. The physiological role or existence of the dimer-of-dimer state is unclear to me, as the bacterial dNTP pools would likely maintain the overall complex in a trimer-of-dimer state. The authors should discuss why the tetrameric structure would be physiologically relevant.

RE: We sincerely thank the reviewer for this insightful comment. The cryo-EM structure of DRT9, determined in the absence of dNTPs, was predominantly tetramer, albeit with a small fraction (~13%) of particles corresponding to hexamer DRT9. However, when 1 mM dNTP was incubated with DRT9, the population shifted to 40% hexamer and 60% tetramer (**Fig. R3**). This suggests a dNTP concentration-dependent dynamic equilibrium between the tetrameric and hexameric states. This transition was further supported by native PAGE, which showed a progressive shift from tetramer to hexamer with increasing dNTP concentrations (**Fig. R5**). Notably, bacterial dNTP pool concentrations typically range from 50 to 200 μM (**ref 1**), which are insufficient to stabilize the overall complex in a trimer-of-dimer state, as evidenced by our PAGE results (**Fig. R5**). We will design more experiments, such as SEC-MALS assay at varying dNTP concentrations, and include the new data and analysis in our revision. Together, we hope our upcoming experiments and the available literature could address the reviewer's concern.

Fig. R5 Native PAGE analyses of DRT9 with increasing dNTP concentrations.

7. DRT9-mediated cDNA synthesis is one of the most exciting findings in this study (Figure S11). If indeed there was no sequence specificity in the cDNA, this could suggest cDNA-mediated regulation in this family of proteins. Could the author please provide the density for the covalent adjunct between the DRT9 C-terminal residues (Ser/Tyr) and cDNA? Additionally, I would suggest that Figure S11 be moved to the main figures as it is one of the impactful observations in this study. Furthermore, I would insist on performing protein sequence alignment among multiple DRT9 homologs to identify the critical amino acid involved in priming reaction. Analyzing the growth curve of priming crucial mutants might offer functional insights if DRT9 is abortive in nature and if the random cDNA serves as a regulatory mechanism in this scenario.

RE: Figure S11 will be moved to the main figures as suggested by the reviewer. We will perform a sequence alignment of DRT9 homologs to identify conserved residues critical for the priming reaction and discuss their functional roles. Growth curves of mutations in these residues under phage infection will also be performed to evaluate their importance for DRT9 activity. These updates will be included in the revised manuscript. Moreover, C-terminal residues were built in the cryo-EM density, whereas the density of the covalent adjunct was not traced, potentially due to poor density in the C-terminal region. The inherent flexibility of the C-terminal loop might facilitate cDNA extension, a mechanism similar to that observed in Abi RTs (ref 2).

8. Please note that in the alignment depicted in Figure S9, there's an extra N-terminal His tag sequence that might be disrupting the alignment. Removing this tag could potentially enhance the alignment features.

RE: We will remove this tag sequence and provide an updated sequence alignment in the revised manuscript.

9. The manuscript discussion abruptly ends. It should be resumed to discuss how the current manuscript advances our understanding of DRTs and establishes a solid foundation for a distinct mechanism. It should also address unanswered futuristic questions that arise from this study.

RE: We will expand the discussion section to incorporate these points as suggested.

Here are some minor but important points:

Please label the organisms' names in the schematics shown in Figure 1b.

Re: We will label the organisms' names in Figure 1b in revision.

What do the authors mean by "The gel is representative of three biological replicates" (Figure 1c legend). This is an in vitro assay, please fix this statement.

Re: The gel shown is representative of three independent experiments. We will modify accordingly.

Please add pixel size to the legend of Figure 1d.

Re: We will add pixel size to the legend of Figure 1d in revision as suggested.

Please mention the sequence similarity between DRT2 and DRT9 in the text.

Re: We will mention the sequence similarity between DRT2 and DRT9 in revision.

Could you please make the PFU assays more visually clear throughout the manuscript? These are hard to see.

Re: We will modify these figures in revision.

There's a typo in Figure 3d and S8a. It should be "assembly".

Re: We will revise these figures as suggested in revision.

In Table 1, under the heading "Post-Refinement," what does the author mean by "model resolution"? How was this calculated?

Re: We evaluated the model using *comprehensive validation* in PHENIX yielding the model resolution. The "resolution" of a model indicates how accurately it represents the details found in the experimental map. A model with higher resolution typically provides a better fit to the map (**ref 3**).

The title of the section "ncRNA conservative analysis" should be corrected.

Re: Thanks, we will revise this title.

In the "In vitro reverse transcription assay" section of the Material and Methods, please provide the percentage of Urea PAGE.

Re: We will add the percentage of Urea PAGE as suggested.

Grammatical errors, superscripts, and other typos need to be corrected throughout the manuscript.

Re: We will carefully revise the whole manuscript as suggested.

Referee #2 (Report for Author)

In this manuscript, Han et al. present a structural and functional investigation of the DRT9 anti-phage defense system, which consists of a reverse transcriptase (RT) and a non-coding RNA (ncRNA). Using cryo-EM, the authors determined molecular structures of the DRT9 RT-ncRNA complex, which adopts distinct conformational and oligomeric states depending on the presence of the substrate, dNTPs. Based on these structural observations combined with functional assays, the authors propose that the dNTP binding induces a conformational rearrangement and an oligomeric transition, ultimately leading to the exposure of the active site and catalytic activation of the RT in the DRT9 system.

This study is of considerable significance, as it provides novel structural insights into the DRT9 system and proposes a unique activation mechanism supported by the structural and biochemical data. The cryo-EM reconstructions are of high quality and contribute valuable information to the field. However, several aspects of the data interpretation remain somewhat ambiguous, and additional analyses or discussion would significantly enhance the clarity and overall impact of the manuscript.

RE: We sincerely appreciate the reviewer's encouraging comments.

Major concerns

1. The cryo-EM data clearly show that the DRT9 RT-ncRNA complex forms tetramers in the absence of dNTPs while transitioning into hexamers upon dNTP binding. Based on this observation, the authors propose that this tetramer-to-hexamer transition is essential for enzymatic activation and the anti-phage defense function of DRT9. However, the precise mechanism underlying this transition is not well explained. Does substrate binding cause dissociation of tetramers into dimers, which subsequently assemble into hexamers? Were such intermediate dimers observed upon dNTP addition during the cryo EM or other experiments?

Alternatively, is there a pre-existing equilibrium between tetramers and dimers in the absence of dNTPs? The SEC data shown in Figure S1b (without dNTPs) suggest a monodisperse population, which appears consistent with a single, likely tetrameric state. The authors are encouraged to explore these possibilities and, if necessary, provide additional data or discussion to clarify the mechanistic details of this

oligomeric transition.

RE: We thank the reviewer for the valuable comments. We did not observe any classes of particles representing the DRT9 dimer during EM data processing of apo and dNTP-bound DRT9. Moreover, our SEC data shown in Figure S1b indicates that the protein predominantly exists as a tetramer in solution. Please find our response to Reviewer #1 major point 2. We will use AUC or SEC-MALS to further characterize the oligomerization status in revision.

As the reviewer highlighted, a dynamic equilibrium indeed exists between tetramer and hexamer, which is likely to be regulated by dNTPs. Please find our response to Reviewer #1 major point 6. We will include these findings in the revised manuscript.

2. In addition to the cryo EM analysis, the authors performed biochemical experiments, in which the oligomeric state of the DRT system could be estimated. It would be helpful to clarify whether the oligomeric states observed in these biochemical assays are fully consistent with the cryo-EM results. For example, in Figure S1b, the authors indicate a peak corresponding to "DRT9 polymer ~450 kDa" but do not explicitly assign this to a specific oligomeric state. Similarly, in Figure S6b, the band shifts observed upon dNTP addition are simply labeled as "DRT9" and "DRT9 oligomer," without clear identification of their stoichiometry. This information would strengthen the mechanistic interpretation. Additionally, the authors should consider investigating whether the loss-of-function mutants, examined throughout the manuscript, show defects in oligomeric assembly. Such analyses would provide further insight into the functional relevance of the mutated residues.

RE: We apologize for the confusion. The oligomeric states observed in these biochemical assays are consistent with our cryo-EM results. We will indicate and clarify these information in our revised manuscript. Furthermore, analytical ultracentrifugation (AUC) or SEC-MALS will be utilized to characterize the oligomerization status for WT and loss-of-function mutants in our revision.

3. It would also be helpful to include a quantitative discussion on the dNTP concentration required for DRT9 activation, considering that dNTPs are constitutively present in bacterial cells. Is there a threshold dNTP concentration necessary to trigger activation? In Figure 6, the authors refer to a "Signal", presumably representing phage-derived factors or secondary signaling molecules that initiate the anti-phage response. Could this "signal" be linked to dNTP-mediated activation? Discussing the potential interplay between these factors would strengthen the biological relevance of the proposed mechanism.

RE: We agree and thank the reviewer for the thoughtful advice. There is indeed a

threshold dNTP concentration necessary to trigger activation. Please refer to our response to reviewer #1 (major point 6) for details. We will include more experimental data, such as varying dNTP concentration for reverse transcription, in our revision.

The signal could be linked to dNTP-mediated activation. One possible scenario is that phages encode proteins that inhibit bacterial dNTPase, thereby increasing dNTP levels to facilitate their own replication (**ref 4-5**). Alternatively, it has been documented that phage proteins also hijack key components for dNTP synthesis to promote dNTP production (**ref 6**). We will include these discussions in our revision.

Minor points

1. (Page 5, Lines 16-17) The authors describe that "DRT2 generates long cDNA product, typically larger than 450 nt. If this observation comes from a previous study, a proper reference should be provided. Additionally, it would be helpful to include a DNA size ladder in Figure 1C for clarity.

Re: We thank the reviewer for the suggestion. We will provide the proper reference and add a DNA size ladder in Figure 1C in revision.

2. (Page 6, Line 2) There appears to be inconsistency in the length of the ncRNA in the DRT9 system-listed as 188 nt in the text, but shown as 180 nt in Figure 1b. This discrepancy should be clarified.

Re: We apologize for the confusion, and we will fix Figure 1b in revision.

3. (Page 9, Line 24) It would enhance the clarity of the structural data if the cryo-EM densities for the bound dNTPs and cDNA products were shown explicitly in a figure panel.

Re: Thanks, we will provide the EM density of bound dNTPs and cDNA products in revision.

References:

1. Buckstein, M.H., He, J. and Rubin, H., 2008. Characterization of nucleotide pools as a function of physiological state in Escherichia coli. *Journal of bacteriology*, 190(2), pp.718-726.
2. Gapińska, M., Zajko, W., Skowronek, K., Figiel, M., Krawczyk, P.S., Egorov, A.A.,

Dziembowski, A., Johansson, M.J. and Nowotny, M., 2024. Structure-functional characterization of Lactococcus AbiA phage defense system. *Nucleic Acids Research*, 52(8), pp.4723-4738.

3. Afonine, P.V., Klaholz, B.P., Moriarty, N.W., Poon, B.K., Sobolev, O.V., Terwilliger, T.C., Adams, P.D. and Urzhumtsev, A., 2018. New tools for the analysis and validation of cryo-EM maps and atomic models. *Biological Crystallography*, 74(9), pp.814-840.

4. Center, M.S., Studier, F.W. and Richardson, C.C., 1970. The structural gene for a T7 endonuclease essential for phage DNA synthesis. *Proceedings of the National Academy of Sciences*, 65(1), pp.242-248.

5. Klemm, B.P., Singh, D., Smith, C.E., Hsu, A.L., Dillard, L.B., Krahn, J.M., London, R.E., Mueller, G.A., Borgnia, M.J. and Schaaper, R.M., 2022. Mechanism by which T7 bacteriophage protein Gp1. 2 inhibits Escherichia coli dGTPase. *Proceedings of the National Academy of Sciences*, 119(37), p.e2123092119.

6. Shen, R., Olcott, M.C., Kim, J., Rajagopal, I. and Mathews, C.K., 2004. Escherichia coli nucleoside diphosphate kinase interactions with T4 phage proteins of deoxyribonucleotide synthesis and possible regulatory functions. *Journal of Biological Chemistry*, 279(31), pp.32225-32232.

Dr. Heng Zhang
TianJin Medical University
No22 Qixiangtai Rd
TianJin 300070
China

22nd Apr 2025

Re: EMBOJ-2025-120298
Non-coding RNA mediates the Defense-Associated Reverse Transcriptase anti-phage oligomerization transition

Dear Dr. Zhang,

Thank you for your tentative point-by-point responses to the comments of our two expert referees. I have now had a chance to go through your proposal for how you would address them, and concluded that your answers appear to be overall reasonable and helpful for clarifying the key concerns of the referees. I would therefore be happy to invite a revision of the study along the lines proposed in your response letter, and to consider it further for EMBO Journal publication. As mentioned previously, crucial points will be better dissection of the different oligomeric forms and their occurrence; and further investigation of cDNA synthesis by DRT9.

Please keep in mind that it is our policy to allow only a single round of (major) revision, and please update me should there be any unexpected problems with the revisions, or should you require an extension beyond the default 3-months deadline. As always, competing manuscript published elsewhere during the course of this revision will not affect our final decision on your study. Finally, please note the detailed information and guidelines on how to prepare a revision below (and in our online Guide to Authors) - closely adhering to them shall greatly facilitate the editorial process at the time of resubmission.

Thank you again for the opportunity to consider this work, and I look forward to receiving your revision in due time.

With kind regards,

Hartmut Vodermaier

9) To facilitate reproducibility and cross-laboratory adoption of methodologies, please structure the Materials & Methods section as outlined in our guide to authors, including a completed Reagents and Tools Table that can be downloaded from our author guidelines as well (<https://www.embopress.org/page/journal/14602075/authorguide#structuredmethods>).

10) Digital image enhancement is acceptable practice, as long as it accurately represents the original data and conforms to community standards. If a figure has been subjected to significant electronic manipulation, this must be clearly noted in the figure legend and/or the 'Materials and Methods' section. The editors reserve the right to request original versions of figures and the original images that were used to assemble the figure. Finally, we generally encourage uploading of numerical as well as gel/blot image source data; for details see: embopress.org/page/journal/14602075/authorguide#sourcedata

In the interest of ensuring the conceptual advance provided by the work, we recommend submitting a revision within 3 months (21st Jul 2025). Please discuss the revision progress ahead of this time with the editor if you require more time to complete the revisions. Use the link below to submit your revision:

Link Not Available

Referee #1:

In the manuscript titled "Non-coding RNA mediates the Defense-Associated Reverse Transcriptase anti-phage oligomerization transition," the authors systematically and functionally characterized an Escherichia coli DRT9 system. They demonstrated that both the reverse transcriptase and non-coding RNA are crucial for the anti-phage function. To investigate its biochemical activity, the authors showed that DRT9 is involved in cDNA synthesis, which is hypothesized to form a covalent bond through the utilization of the C-terminal tail of the reverse transcriptase protein. Furthermore, using cryogenic electron microscopy, the authors determined two cryoEM structures of the DRT9 complex, one with and one without the substrate. Finally, the authors proposed a regulatory mechanism. They suggested that upon substrate binding, the DRT9 complex undergoes a transition, where the dimer-of-dimer complex changes to a trimer-of-dimer complex.

The authors present a comprehensive structural analysis of DRT9 complexes, examining two distinct forms. They delve into the structural changes and propose an intriguing model that suggests how the DRT9 system facilitates the anti-phage function. While the structural and functional characterization is meticulously detailed, the proposed model lacks compelling experimental evidence and a clear rationale for considering the dimer-of-dimer state as physiologically relevant. Below are my major concerns and suggestions aimed at enhancing the quality of this manuscript, and I hope the authors find them helpful.

Major points:

The authors determined the sequence of cDNA from their in vitro reconstituted reverse transcription reactions. However, they did not discuss the mapping of these sequence reads in this manuscript. Do these map to non-coding RNA (ncRNA) sequence? If not, are they similar to ncRNA-expressing plasmid regions? Understanding what the synthesized cDNA codes for (if it does) is crucial to understanding the mechanism of DRT9. At the very least, the authors should submit and share their sequencing data in a publicly accessible repository and provide an explanation for their inability to map these reads.

Could the authors please clarify how the size of the DRT9 polymer ("~450 kDa") was determined in Figure S1b? Were protein standards used for this calculation, or is it based on multi-angle light scattering or other techniques?

The "Material and Methods" section is missing crucial information about how the DRT2 was expressed, purified, and the CryoEM data collection for 2D classes in Figure 1d. The author should either provide this information or cite the appropriate sources.

To understand the functional importance of tetrameric assembly, the authors aimed at disrupting the oligomer using ncRNA mutagenesis (G114C) and by deleting the assembly loop region 288-299. These alterations were analyzed using native PAGE analysis, but the results don't support the claim stated "Deletion of the assembly loop (D288-299) disrupts the higher-order oligomerization of DRT9." For instance, the wild-type complex, which forms a tetramer in the absence of dNTP, exhibited similar gel shift patterns to the two mutants studied. Could the author clarify why these mutants and the wild-type complex move at the same size? Alternatively, could these discrepancies be resolved by employing a size exclusion-based assay, as demonstrated for the wild-type complex in Figure S1b. The current characterization of these mutants does not support the disruption of tetrameric assembly.

The authors employed D3 symmetry during the cryoEM data processing of the substrate-bound DRT9 complex. Could they have observed the cDNA density in all the protomers before processing as a D3 complex? Furthermore, did the authors attempt performing symmetry expansion to enhance the structure resolution further? Could the authors show the additional cryoEM density they mentioned for the cDNA product? Perhaps symmetry expansion could aid in resolving the cDNA density to a higher resolution. Lastly, it remains unclear why a significant portion of particles (77.8%) were excluded from further data processing in Figure S7. An explanation for this exclusion should be provided.

The authors showed a dimer-of-dimer to trimer-of-dimer transitions in DRT9 in the presence of dNTP mix. The physiological role or existence of the dimer-of-dimer state is unclear to me, as the bacterial dNTP pools would likely maintain the overall complex in a trimer-of-dimer state. The authors should discuss why the tetrameric structure would be physiologically relevant.

DRT9-mediated cDNA synthesis is one of the most exciting findings in this study (Figure S11). If indeed there was no sequence specificity in the cDNA, this could suggest cDNA-mediated regulation in this family of proteins. Could the author please provide the density for the covalent adjunct between the DRT9 C-terminal residues (Ser/Tyr) and cDNA? Additionally, I would suggest that Figure S11 be moved to the main figures as it is one of the impactful observation in this study. Furthermore, I would insist on performing protein sequence alignment among multiple DRT9 homologs to identify the critical amino acid involved in priming reaction. Analyzing the growth curve of priming crucial mutants might offer functional insights if DRT9 is abortive in nature and if the random cDNA serves as a regulatory mechanism in this scenario.

Please note that in the alignment depicted in Figure S9, there's an extra N-terminal His tag sequence that might be disrupting the alignment. Removing this tag could potentially enhance the alignment features.

The manuscript discussion abruptly ends. It should be resumed to discuss how the current manuscript advances our understanding of DRTs and establishes a solid foundation for a distinct mechanism. It should also address unanswered futuristic questions that arise from this study.

Here are some minor but important points:

Please label the organisms' names in the schematics shown in Figure 1b.

What do the authors mean by "The gel is representative of three biological replicates" (Figure 1c legend). This is an in vitro assay, please fix this statement.

Please add pixel size to the legend of Figure 1d.

Please mention the sequence similarity between DRT2 and DRT9 in the text.

Could you please make the PFU assays more visually clear throughout the manuscript? These are hard to see.

There's a typo in Figure 3d and S8a. It should be "assembly".

In Table 1, under the heading "Post-Refinement," what does the author mean by "model resolution"? How was this calculated?

The title of the section "ncRNA conservative analysis" should be corrected.

In the "In vitro reverse transcription assay" section of the Material and Methods, please provide the percentage of Urea PAGE.

Grammatical errors, superscripts, and other typos need to be corrected throughout the manuscript.

Referee #2:

In this manuscript, Han et al. present a structural and functional investigation of the DRT9 anti-phage defense system, which consists of a reverse transcriptase (RT) and a non-coding RNA (ncRNA). Using cryo-EM, the authors determined molecular structures of the DRT9 RT-ncRNA complex, which adopts distinct conformational and oligomeric states depending on the presence of the substrate, dNTPs. Based on these structural observations combined with functional assays, the authors propose that the dNTP binding induces a conformational rearrangement and an oligomeric transition, ultimately leading to the exposure of the active site and catalytic activation of the RT in the DRT9 system.

This study is of considerable significance, as it provides novel structural insights into the DRT9 system and proposes a unique activation mechanism supported by the structural and biochemical data. The cryo-EM reconstructions are of high quality and contribute valuable information to the field. However, several aspects of the data interpretation remain somewhat ambiguous, and additional analyses or discussion would significantly enhance the clarity and overall impact of the manuscript.

Major concerns

1. The cryo-EM data clearly show that the DRT9 RT-ncRNA complex forms tetramers in the absence of dNTPs while transitioning into hexamers upon dNTP binding. Based on this observation, the authors propose that this tetramer-to-hexamer transition is essential for enzymatic activation and the anti-phage defense function of DRT9. However, the precise mechanism underlying this transition is not well explained. Does substrate binding cause dissociation of tetramers into dimers, which subsequently assemble into hexamers? Were such intermediate dimers observed upon dNTP addition during the cryo EM or other experiments? Alternatively, is there a pre-existing equilibrium between tetramers and dimers in the absence of dNTPs? The SEC data shown in Figure S1b (without dNTPs) suggest a monodisperse population, which appears consistent with a single, likely tetrameric state. The authors are encouraged to explore these possibilities and, if necessary, provide additional data or discussion to clarify the mechanistic details of this oligomeric transition.

2. In addition to the cryo EM analysis, the authors performed biochemical experiments, in which the oligomeric state of the DRT system could be estimated. It would be helpful to clarify whether the oligomeric states observed in these biochemical assays are fully consistent with the cryo-EM results. For example, in Figure S1b, the authors indicate a peak corresponding to "DRT9 polymer ~450 kDa" but do not explicitly assign this to a specific oligomeric state. Similarly, in Figure S6b, the band shifts observed upon dNTP addition are simply labeled as "DRT9" and "DRT9 oligomer," without clear identification of their stoichiometry. This information would strengthen the mechanistic interpretation. Additionally, the authors should consider investigating whether the loss-of-function mutants, examined throughout the manuscript, show defects in oligomeric assembly. Such analyses would provide further insight into the functional relevance of the mutated residues.

3. It would also be helpful to include a quantitative discussion on the dNTP concentration required for DRT9 activation, considering that dNTPs are constitutively present in bacterial cells. Is there a threshold dNTP concentration necessary to trigger activation? In Figure 6, the authors refer to a "Signal", presumably representing phage-derived factors or secondary signaling molecules that initiate the anti-phage response. Could this "signal" be linked to dNTP-mediated activation? Discussing the potential interplay between these factors would strengthen the biological relevance of the proposed mechanism.

Minor points

1. (Page 5, Lines 16-17) The authors describe that "DRT2 generates long cDNA product, typically larger than 450 nt. If this observation comes from a previous study, a proper reference should be provided. Additionally, it would be helpful to include a DNA size ladder in Figure 1C for clarity.

2. (Page 6, Line 2) There appears to be inconsistency in the length of the ncRNA in the DRT9 system-listed as 188 nt in the text, but shown as 180 nt in Figure 1b. This discrepancy should be clarified.

3. (Page 9, Line 24) It would enhance the clarity of the structural data if the cryo-EM densities for the bound dNTPs and cDNA products were shown explicitly in a figure panel.

Response to Reviewer Comments

We thank the reviewers for their positive and insightful comments, which are invaluable in improving our manuscript. Below we have addressed the specific points raised by the reviewers and amended our manuscript accordingly. The revisions have been marked in blue font in the revised manuscript. We hope the reviewers find it suitable for publication.

Referee #1 (Report for Author)

In the manuscript titled "Non-coding RNA mediates the Defense-Associated Reverse Transcriptase anti-phage oligomerization transition," the authors systematically and functionally characterized an Escherichia coli DRT9 system. They demonstrated that both the reverse transcriptase and non-coding RNA are crucial for the anti-phage function. To investigate its biochemical activity, the authors showed that DRT9 is involved in cDNA synthesis, which is hypothesized to form a covalent bond through the utilization of the C-terminal tail of the reverse transcriptase protein. Furthermore, using cryogenic electron microscopy, the authors determined two cryoEM structures of the DRT9 complex, one with and one without the substrate. Finally, the authors proposed a regulatory mechanism. They suggested that upon substrate binding, the DRT9 complex undergoes a transition, where the dimer-of-dimer complex changes to a trimer-of-dimer complex.

The authors present a comprehensive structural analysis of DRT9 complexes, examining two distinct forms. They delve into the structural changes and propose an intriguing model that suggests how the DRT9 system facilitates the anti-phage function. While the structural and functional characterization is meticulously detailed, the proposed model lacks compelling experimental evidence and a clear rationale for considering the dimer-of-dimer state as physiologically relevant. Below are my major concerns and suggestions aimed at enhancing the quality of this manuscript, and I hope the authors find them helpful.

RE: We appreciate the reviewer's constructive feedback, which has helped us strengthen the manuscript.

Major points:

1. The authors determined the sequence of cDNA from their in vitro reconstituted reverse transcription reactions. However, they did not discuss the mapping of these sequence reads in this manuscript. Do these map to non-coding RNA (ncRNA) sequence? If not, are they similar to ncRNA-expressing plasmid regions? Understanding what the synthesized cDNA codes for (if it does) is crucial to understanding the mechanism of DRT9. At the very least, the authors should submit

and share their sequencing data in a publicly accessible repository and provide an explanation for their inability to map these reads.

RE: We appreciate the reviewer's insightful comment. In the revised manuscript, we characterized the synthesized cDNA codes. Although the sequence of the synthesized cDNA products failed to map to either ncRNAs or the *E. coli* genome, we observed a high abundance of poly-A motif sequences among unmapped reads (Figure 1E). Furthermore, in vitro transcription analysis suggests that the DRT9 system can efficiently generate cDNA products in the presence of dATP only, but not in the presence of the other three dNTPs (Figure 1F, Appendix Figure S1B). Following the reviewer's suggestion, our next-generation sequencing data has been submitted to the NCBI SRA database (BioProject: PRJNA1280644).

2. Could the authors please clarify how the size of the DRT9 polymer ("~450 kDa") was determined in Figure S1b? Were protein standards used for this calculation, or is it based on multi-angle light scattering or other techniques?

RE: We apologize for the confusion. The molecular weight of the DRT9 polymer was estimated using size-exclusion chromatography (SEC) with protein standards (Fig. R1, Appendix Figure S1D). The theoretical molecular weights of the tetrameric and hexameric DRT9 complexes are ~ 450 kDa and 690 kDa, respectively, which are consistent with their experimental data. We have clarified this description in the revised manuscript.

Fig. R1 Left: Elution profiles of DRT9 and standard makers. Right: Molecular weight calibration curve

3. The "Material and Methods" section is missing crucial information about how the DRT2 was expressed, purified, and the CryoEM data collection for 2D classes in Figure 1d. The author should either provide this information or cite the appropriate sources.

RE: Thanks for pointing this out. We have added this information in "Methods" section in the revised manuscript.

4. To understand the functional importance of tetrameric assembly, the authors aimed at disrupting the oligomer using ncRNA mutagenesis (G114C) and by region 288-299.

These alterations were analyzed using native PAGE analysis, but the results don't support the claim stated "Deletion of the assembly loop (D288-299) disrupts the higher-order oligomerization of DRT9." For instance, the wild-type complex, which forms a tetramer in the absence of dNTP, exhibited similar gel shift patterns to the two mutants studied. Could the author clarify why these mutants and the wild-type complex move at the same size? Alternatively, could these discrepancies be resolved by employing a size exclusion-based assay, as demonstrated for the wild-type complex in Figure S1b. The current characterization of these mutants does not support the disruption of tetrameric assembly.

RE: Guided by the reviewer's valuable suggestion, we employed size exclusion chromatography assay to assess the oligomeric state of these DRT9 mutants. The SEC results indicate that mutations in the C94-G114 Watson-Crick base pairs and the assembly loop region (residues 288-299) disrupt the tetrameric assembly of DRT9, demonstrating their critical roles in tetramer formation (**Appendix Figure S6B**). We have added these SEC results in the revised manuscript for clarity.

5. The authors employed D3 symmetry during the cryoEM data processing of the substrate-bound DRT9 complex. Could they have observed the cDNA density in all the protomers before processing as a D3 complex? Furthermore, did the authors attempt performing symmetry expansion to enhance the structure resolution further? Could the authors show the additional cryoEM density they mentioned for the? Perhaps symmetry expansion could aid in resolving the cDNA density to a higher resolution. Lastly, it remains unclear why a significant portion of particles (77.8%) were excluded from further data processing in Figure S7. An explanation for this exclusion should be provided.

RE: As suggested by the reviewer, we re-processed the cryo-EM data without imposing symmetry (C1 symmetry) (**Appendix Figure S8**) and confirmed the presence of cDNA density in the substrate-bound complex (**Figure 5A**), which is consistent with our previous model. Following the reviewer's advice, we also performed symmetry expansion, however, the resolution obtained with symmetry expansion (3.41 Å) is lower than that obtained with D3 symmetry (3.0 Å) (**Fig. R2**). The dATP/cDNA densities are not clearly observed in the 77.8% of hexamer particles. Therefore, we did not pursue additional data processing. We have included this information in the revised manuscript (Methods section, "Cryo-EM data processing").

Fig. R2 Symmetry expansion was applied on the refined particles.

6. The authors showed a dimer-of-dimer to trimer-of-dimer transitions in DRT9 in the presence of dNTP mix. The physiological role or existence of the dimer-of-dimer state is unclear to me, as the bacterial dNTP pools would likely maintain the overall complex in a trimer-of-dimer state. The authors should discuss why the tetrameric structure would be physiologically relevant.

RE: We sincerely thank the reviewer for this insightful comment. The cryo-EM structure of DRT9, determined in the absence of dNTPs, was predominantly tetramer, albeit with a small fraction (~13%) of particles corresponding to hexamer DRT9 (**Appendix Figure S2**). However, when dNTP was incubated with DRT9, the hexameric population remarkably shifted to ~40% (**Appendix Figure S8**). This suggests a dNTP concentration-dependent dynamic equilibrium between the tetrameric and hexameric states. This transition was further supported by native PAGE, which showed a progressive shift from tetramer to hexamer with increasing dNTP concentrations (**Appendix Figure S7**). Notably, bacterial dNTP pool concentrations, particularly the level of dATP, typically range from 20 to 100 μM (**ref 1-3**), which are insufficient to induce the overall complex to transition into the catalytically active hexameric state (**Appendix Figure S7**). It's also worth noting that dNTP levels decrease during bacterial growth (**ref 1**). Therefore, it is plausible that DRT9 remains in an inactive tetrameric state under physiological dNTP level and becomes activated when the dNTP concentration exceeds a critical threshold. We have included these points in the revised manuscript (paragraph 2 of Discussion section).

7. DRT9-mediated cDNA synthesis is one of the most exciting findings in this study

(Figure S11). If indeed there was no sequence specificity in the cDNA, this could suggest cDNA-mediated regulation in this family of proteins. Could the author please provide the density for the covalent adjunct between the DRT9 C-terminal residues (Ser/Tyr) and cDNA? Additionally, I would suggest that Figure S11 be moved to the main figures as it is one of the impactful observations in this study. Furthermore, I would insist on performing protein sequence alignment among multiple DRT9 homologs to identify the critical amino acid involved in priming reaction. Analyzing the growth curve of priming crucial mutants might offer functional insights if DRT9 is abortive in nature and if the random cDNA serves as a regulatory mechanism in this scenario.

RE: We are sincerely grateful to the reviewer for the thoughtful suggestion, which helps us draw more impactful conclusions. We have moved Figure S11 to the main **Figure 6** as suggested. We also performed a sequence alignment of DRT9 homologs and identified two conserved C-terminal tyrosine residues potentially involved in the priming reaction (**Figure 6A and Appendix Figure S12B**). Additionally, we conducted growth curve analyses of these critical residue tyrosine mutants under phage infection, demonstrating their importance in DRT9-mediated defense (**Figure 6B**).

Regarding the covalent linkage between the C-terminal residues and cDNA, we could not observe clear cryo-EM density corresponding to such a covalent adjunct, likely due to the intrinsic flexibility of the C-terminal loop, which may facilitate cDNA elongation, a mechanism proposed for Abi RTs (**ref 4**). We have included this point in the revised manuscript (paragraph 3 of Discussion section).

8. Please note that in the alignment depicted in Figure S9, there's an extra N-terminal His tag sequence that might be disrupting the alignment. Removing this tag could potentially enhance the alignment features.

RE: Thanks for the helpful suggestion. We have removed extra N-terminal His tag sequence and provided an updated sequence alignment in the revised manuscript (now **Appendix Figure S10**).

9. The manuscript discussion abruptly ends. It should be resumed to discuss how the current manuscript advances our understanding of DRTs and establishes a solid foundation for a distinct mechanism. It should also address unanswered futuristic questions that arise from this study.

RE: Thank you for these helpful suggestions. We have expanded the discussion section to incorporate the points as suggested, hoping to address the reviewer's concerns.

Here are some minor but important points:

Please label the organisms' names in the schematics shown in Figure 1b.

Re: We have labeled the organisms' names in revision.

What do the authors mean by "The gel is representative of three biological replicates" (Figure 1c legend). This is an in vitro assay, please fix this statement.

Re: Our apologies for the confusion. The gel shown is representative of three independent experiments. We have modified the text in the revised manuscript.

Please add pixel size to the legend of Figure 1d.

Re: We have added pixel size to the legend in revision as suggested (now **Appendix Figure S1C**)

Please mention the sequence similarity between DRT2 and DRT9 in the text.

Re: We have mentioned the sequence similarity between DRT2 and DRT9 in revision (paragraph 1 of Results section).

Could you please make the PFU assays more visually clear throughout the manuscript? These are hard to see.

Re: We have included the updated PFU assays in the revised manuscript.

There's a typo in Figure 3d and S8a. It should be "assembly".

Re: Thanks for pointing these out. We have revised these figures as suggested in the revision (now **Figure 3D and Appendix Figure S9A**).

In Table 1, under the heading "Post-Refinement," what does the author mean by "model resolution"? How was this calculated?

Re: We evaluated the model using *comprehensive validation* in PHENIX, yielding the model resolution. The "resolution" of a model indicates how accurately it represents the details found in the experimental map. A model with higher resolution typically provides a better fit to the map (**ref 5**).

The title of the section "ncRNA conservative analysis" should be corrected.

Re: Thanks, we have revised this title.

In the "In vitro reverse transcription assay" section of the Material and Methods, please provide the percentage of Urea PAGE.

Re: We have added the percentage of Urea PAGE as suggested.

Grammatical errors, superscripts, and other typos need to be corrected throughout the manuscript.

Re: Thanks for pointing these out. We have carefully revised the whole manuscript as suggested.

Referee #2 (Report for Author)

In this manuscript, Han et al. present a structural and functional investigation of the DRT9 anti-phage defense system, which consists of a reverse transcriptase (RT) and a

non-coding RNA (ncRNA). Using cryo-EM, the authors determined molecular structures of the DRT9 RT-ncRNA complex, which adopts distinct conformational and oligomeric states depending on the presence of the substrate, dNTPs. Based on these structural observations combined with functional assays, the authors propose that the dNTP binding induces a conformational rearrangement and an oligomeric transition, ultimately leading to the exposure of the active site and catalytic activation of the RT in the DRT9 system.

This study is of considerable significance, as it provides novel structural insights into the DRT9 system and proposes a unique activation mechanism supported by the structural and biochemical data. The cryo-EM reconstitutions are of high quality and contribute valuable information to the field. However, several aspects of the data interpretation remain somewhat ambiguous, and additional analyses or discussion would significantly enhance the clarity and overall impact of the manuscript.

RE: We are grateful to the reviewer for the encouraging comments that have improved our manuscript.

Major concerns

1. The cryo-EM data clearly show that the DRT9 RT-ncRNA complex forms tetramers in the absence of dNTPs while transitioning into hexamers upon dNTP binding. Based on this observation, the authors propose that this tetramer-to-hexamer transition is essential for enzymatic activation and the anti-phage defense function of DRT9.

However, the precise mechanism underlying this transition is not well explained.

Does substrate binding cause dissociation of tetramers into dimers, which subsequently assemble into hexamers? Were such intermediate dimers observed upon dNTP addition during the cryo EM or other experiments?

Alternatively, is there a pre-existing equilibrium between tetramers and dimers in the absence of dNTPs? The SEC data shown in Figure S1b (without dNTPs) suggest a monodisperse population, which appears consistent with a single, likely tetrameric state. The authors are encouraged to explore these possibilities and, if necessary, provide additional data or discussion to clarify the mechanistic details of this oligomeric transition.

RE: We thank the reviewers for the opportunity to clarify these points. During cryo-EM data processing of both the apo and dNTP-bound DRT9 samples (**Appendix Figure S2, S8**), we did not observe any significant particle classes corresponding to dimers or other intermediate oligomeric states except for tetramer and hexamer; the particles primarily correspond to tetramers and exhibit an increased proportion of hexamers upon dNTP binding.

Inspired by the reviewers' thoughtful comments, we re-analyzed the EM dataset and

ran the native PAGE with various dNTPs concentration. These results indeed indicate a dynamic equilibrium between tetrameric and hexameric states, regulated by dNTP concentration (**Appendix Figure S2, S7, S8**). We have included these additional data and expanded discussion in the revised manuscript to clarify this dynamic equilibrium and the mechanism of DRT9 oligomerization transition (paragraph 2 of Discussion section).

2. In addition to the cryo EM analysis, the authors performed biochemical experiments, in which the oligomeric state of the DRT system could be estimated. It would be helpful to clarify whether the oligomeric states observed in these biochemical assays are fully consistent with the cryo-EM results. For example, in Figure S1b, the authors indicate a peak corresponding to "DRT9 polymer ~450 kDa" but do not explicitly assign this to a specific oligomeric state. Similarly, in Figure S6b, the band shifts observed upon dNTP addition are simply labeled as "DRT9" and "DRT9 oligomer," without clear identification of their stoichiometry. This information would strengthen the mechanistic interpretation. Additionally, the authors should consider investigating whether the loss-of-function mutants, examined throughout the manuscript, show defects in oligomeric assembly. Such analyses would provide further insight into the functional relevance of the mutated residues.

RE: We apologize for the confusion. We have clarified the assignment of oligomeric states in the revised manuscript (now **Appendix Figures S1, S6 and S7**).

Additionally, to further explore the functional relevance of critical residues, we have performed gel-filtration analyses to characterize the oligomerization states of wild-type DRT9 and loss-of-function mutants, as shown in **Appendix Figure S12A**. The results show that the loss-of-function mutants do not alter the oligomerization status.

3. It would also be helpful to include a quantitative discussion on the dNTP concentration required for DRT9 activation, considering that dNTPs are constitutively present in bacterial cells. Is there a threshold dNTP concentration necessary to trigger activation? In Figure 6, the authors refer to a "Signal", presumably representing phage-derived factors or secondary signaling molecules that initiate the anti-phage response. Could this "signal" be linked to dNTP-mediated activation? Discussing the potential interplay between these factors would strengthen the biological relevance of the proposed mechanism.

RE: We thank the reviewer for the thoughtful advice. There is indeed a threshold dNTP concentration necessary to trigger activation. We have included more experimental data, such as varying dNTP concentration for reverse transcription, in our revision (**Appendix Figure S1B**). Notably, bacterial dNTP pool concentrations, particularly the level of dATP, typically range from 20 to 100 μM (**ref 1-3**), which are

insufficient to induce the overall complex to transition into the catalytically active hexameric state (**Appendix Figure S7**). It's also worth noting that dNTP levels decrease during bacterial growth (**ref 1**). Therefore, it is plausible that DRT9 remains in an inactive tetrameric state under physiological dNTP level and becomes activated when the dNTP concentration exceeds a critical threshold. Thus, we agree with the reviewers that a signal could be linked to dNTP-mediated activation. One possible scenario is that phages encode proteins that inhibit bacterial dNTPase, thereby increasing dNTP levels to facilitate their own replication (**ref 6-7**). Alternatively, it has been documented that phage proteins also hijack key components for dNTP synthesis to promote dNTP production (**ref 8**). We have included these discussions in our revision (paragraph 2 of Discussion section).

Minor points

1. (Page 5, Lines 16-17) The authors describe that "DRT2 generates long cDNA product, typically larger than 450 nt. If this observation comes from a previous study, a proper reference should be provided. Additionally, it would be helpful to include a DNA size ladder in Figure 1C for clarity.

Re: We thank the reviewer for the suggestion and have provided the proper reference. As suggested, a DNA size ladder was used (**Fig. R3**), however, it is important to note that the relative size does not accurately reflect the authentic size of the product due to the use of Fluorescein-12-dATP in our reverse transcription assay. To avoid the possible confusion, we did not include the DNA ladder in our revised manuscript.

Fig. R3 In vitro reverse transcription assay of DRT2 and DRT9 systems variants

2. (Page 6, Line 2) There appears to be inconsistency in the length of the ncRNA in the DRT9 system-listed as 188 nt in the text, but shown as 180 nt in Figure 1b. This discrepancy should be clarified.

Re: We thank the reviewer for noting this point, and we have fixed Figure 1B in revision.

3. (Page 9, Line 24) It would enhance the clarity of the structural data if the cryo-EM

densities for the bound dNTPs and cDNA products were shown explicitly in a figure panel.

Re: We have provided the EM density of bound dATPs and cDNA products in revision as suggested (**Figure 5A**).

References:

1. Buckstein, M.H., He, J. and Rubin, H., 2008. Characterization of nucleotide pools as a function of physiological state in *Escherichia coli*. *Journal of bacteriology*, 190(2), pp.718-726.
2. Bennett, B.D., Kimball, E.H., Gao, M., Osterhout, R., Van Dien, S.J. and Rabinowitz, J.D., 2009. Absolute metabolite concentrations and implied enzyme active site occupancy in *Escherichia coli*. *Nature chemical biology*, 5(8), pp.593-599.
3. Wheeler, L.J., Rajagopal, I. and Mathews, C.K., 2005. Stimulation of mutagenesis by proportional deoxyribonucleoside triphosphate accumulation in *Escherichia coli*. *DNA repair*, 4(12), pp.1450-1456.
4. Gapińska, M., Zajko, W., Skowronek, K., Figiel, M., Krawczyk, P.S., Egorov, A.A., Dziembowski, A., Johansson, M.J. and Nowotny, M., 2024. Structure-functional characterization of *Lactococcus* AbiA phage defense system. *Nucleic Acids Research*, 52(8), pp.4723-4738
5. Afonine, P.V., Klaholz, B.P., Moriarty, N.W., Poon, B.K., Sobolev, O.V., Terwilliger, T.C., Adams, P.D. and Urzhumtsev, A., 2018. New tools for the analysis and validation of cryo-EM maps and atomic models. *Biological Crystallography*, 74(9), pp.814-840.
6. Center, M.S., Studier, F.W. and Richardson, C.C., 1970. The structural gene for a T7 endonuclease essential for phage DNA synthesis. *Proceedings of the National Academy of Sciences*, 65(1), pp.242-248.
7. Klemm, B.P., Singh, D., Smith, C.E., Hsu, A.L., Dillard, L.B., Krahn, J.M., London, R.E., Mueller, G.A., Borgnia, M.J. and Schaaper, R.M., 2022. Mechanism by which T7 bacteriophage protein Gp1. 2 inhibits *Escherichia coli* dGTPase. *Proceedings of the National Academy of Sciences*, 119(37), p.e2123092119.
8. Shen, R., Olcott, M.C., Kim, J., Rajagopal, I. and Mathews, C.K., 2004. *Escherichia coli* nucleoside diphosphate kinase interactions with T4 phage proteins of deoxyribonucleotide synthesis and possible regulatory functions. *Journal of Biological Chemistry*, 279(31), pp.32225-32232.

Dr. Heng Zhang
TianJin Medical University
Department of Biochemistry and Molecular Biology
No22 Qixiangtai Rd
TianJin 300070
China

6th Aug 2025

Re: EMBOJ-2025-120298R
Non-coding RNA mediates the Defense-Associated Reverse Transcriptase (DRT) anti-phage oligomerization transition

Dear Dr. Zhang,

Thank you for submitting your final revised manuscript for our consideration. I am pleased to inform you that we have now accepted it for publication in The EMBO Journal.

Yours sincerely,

Hartmut Vodermaier

Referee #1:

In the revised version of this manuscript, the authors have addressed all of this reviewer's concerns. They have included new data for several crucial experiments that were essential to substantiate the claims made in this study. Overall, the scientific quality and structure of the manuscript have significantly improved. Therefore, this timely and important contribution should be considered for publication.

Referee #2:

All concerns are properly addressed. I recommend the publication of this manuscript.